

# CIAO observatory main upgrade: building up an ACTRIS compliant aerosol in-situ laboratory

Teresa Laurita[1], Alessandro Mauceri[1], Francesco Cardellicchio[1], Emilio Lapenna[1], Benedetto De Rosa[1], Serena Trippetta[1], Michail Mytilinaios[1], Davide Amodio[1], Aldo Giunta[1], Ermann Ripepi[1], Canio Colangelo[1], Nikolaos Papagiannopoulos[1], Francesca Morrongiello[1], Claudio Dema[1], Simone Gagliardi[1], Carmela Cornacchia[1], Rosa Maria Petracca Altieri[1], Aldo Amodeo[1], Marco Rosoldi[1], Donato Summa[1], Gelsomina Pappalardo[1], Lucia Mona[1]

[1]Consiglio Nazionale delle Ricerche – Istituto di Metodologie per l'Analisi Ambientale CNR-IMAA, C. da S. Loja, Tito Scalo, Potenza, 85050, Italy

*Correspondence to*: Teresa Laurita (teresa.laurita@cnr.it)

**Abstract**

This paper describes the aerosol in-situ laboratory at CIAO (CNR IMAA Atmospheric Observatory) in South Italy, outlining its configuration and detailing each instrument and sampling lines. The CIAO observatory has been collecting observations of atmospheric components since 2000. Initially the activities revolved around aerosol lidar, later radiosounding and cloud remote sensing observations were added over the years and made CIAO a leading atmospheric observatory in the Mediterranean region. In 2018, a significant upgrade started for enhancing the observational capability by adding aerosol in-situ instruments, with the objective to push new research boundaries for aerosol characterization and multi-instrumental synergistic approaches. Here, we describe each technical implementation step for building up an extensive aerosol in-situ laboratory compliant with ACTRIS (Aerosol Clouds and Trace gases Research InfraStructure) standard operating procedures. Starting from scratch, the long path initiated in 2018, with the design of the laboratory in terms of instruments, container organization, inlets and sampling lines optimizations, that required time and interactions with experts in the field. Reporting here all the details about the final solutions implemented at CIAO, this paper will be, for new aerosol in-situ laboratory, a practical guide for the implementation of the aerosol in-situ observational site.

## 1 Introduction

The importance of a quantitative and qualitative assessment of atmospheric aerosol characteristics has been recognized since many years: aerosols are responsible for direct and indirect effects on atmospheric processes, affecting climate and human health, as well as precipitation cycle and air quality (e.g. Pöschl, 2005). Depending on their sources, aerosols appear in different sizes/shapes and their relatively short lifetime makes the physical and chemical properties extremely variable both on temporal





and spatial scales. Because of the inherent complexity of aerosols, a single measurement technique providing all the relevant information is not available: thus, a multi-instrument approach is needed. The combination of different techniques and observational platforms can be crucial for a better understanding of the presence and the characteristics of atmospheric aerosols, as well as their role in the large variety of processes in which they are involved. Aerosol Clouds and Trace Gases Research InfraStructure (ACTRIS, www.actris.eu) is the European Research Infrastructure (RI) aiming to integrate previous existing networks for the characterization of aerosols, clouds and trace gases using and integrating in-situ and remote sensing observations, and experimental platforms for the characterization of atmospheric components under controlled environments. An overarching investigation of the atmosphere which accounts for all these three components is a winning strategy: For instance, aerosols act as cloud condensation nuclei (CNN) affecting the cloud properties and lifetime; emitted gas species may act as precursors to form new particles in the atmosphere, i.e., the secondary aerosol. Integrated approaches of remote sensing and in-situ observations allows to take the most from the detailed and accurate characterization in terms of morphology of particles, dimension and chemical composition: the remote sensing provide the vertical profile of physical and optical properties information which are essential for investigating aerosol layers, long range transportation, mixing processes and aerosol-cloud interactions; the latter is the only approach to provide the chemical composition and reliable data at ground level, where aerosols affect ecosystems and humans.

In this scenario, the CNR-IMAA (Consiglio Nazionale delle Ricerche – Istituto di Metodologie per l'Analisi Ambientale) Atmospheric Observatory (CIAO; Madonna et al., 2011), operating since 2000, has been recently upgraded with the aerosol in-situ observational component, thus complementing the multi-year high-quality aerosol remote sensing data record. The combination of the aerosol in-situ measurements with remote sensing observations is expected to strengthen fundamental knowledge about aerosol impact on human health, ecosystems, and climate. This combination can be achieved either by comparing or complementing the techniques: the results of the comparison will allow to reduce the uncertainty of aerosol measurements in the atmosphere, with a subsequent improvement of model predictions on climate change, whereas the complementarity results in the possibility of investigating the aerosol from the ground up to the stratosphere. The new aerosol in-situ facility at CIAO, funded by the Italian Ministry of University and Research through the PER-ACTRIS-IT project (https://www.imaa.cnr.it/en/projects/38-attivita/progetti/713-per-actris-it, last access: 6 December 2023), has received initial acceptance as ACTRIS National Facility observational platform for the measurements of at least the obligatory ACTRIS aerosol in-situ variables. The site will start the next phase of the labelling process in 2024.

In this paper we present a concise overview of the observatory, focusing on the characteristics of the recently established ACTRIS-compliant in-situ facility, with the main aim to benefit the aerosol community providing a comprehensive and detailed description of technical solutions for the implementation of such component. After a short description of CIAO and typical atmospheric conditions in Sect. 2, Sect. 3 reports about the remote sensing instrumentation currently operating at CIAO. Section 4 represents the core of this paper, providing the in-depth description of the in-situ facility with the detailed configuration of each instrument and sampling lines. Finally, Sect. 5 illustrates three scientific topics to be studied at CIAO with the synergistic deployment of aerosol in-situ and remote sensing measurements.


## 2 Description of the site

Equipped with state-of-the-art systems for remote-sensing and in-situ measurements of aerosol, CIAO (https://ciao.imaa.cnr.it,
last access: 4 December 2023) is currently a reference observatory for atmospheric research in Europe. The site is located on
the Southern Apennine in Italy (Tito Scalo, 40.60° N, 15.72° E, 760 m a.s.l.), in a plain surrounded by low mountains, less
than 150 km away from the West, South and East coasts (Fig. 1).

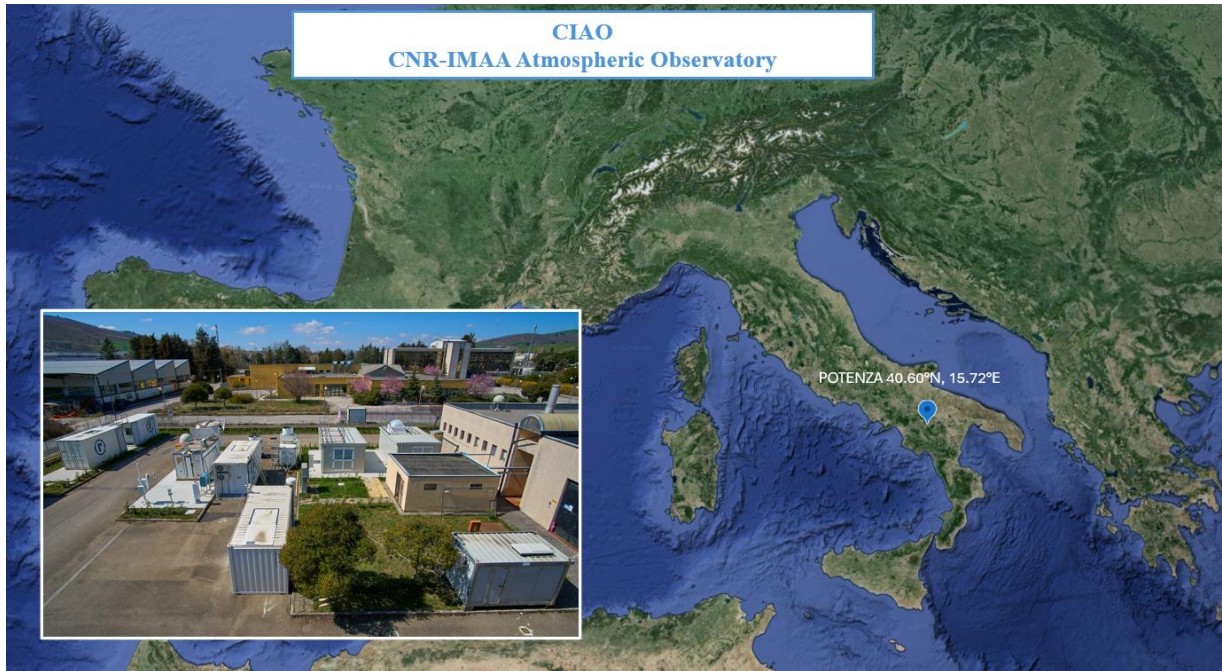

**Figure 1: Location and image of CNR-IMAA Atmospheric Observatory (© Google Earth)**

Therefore, it operates in a typical mountainous weather strongly influenced by Mediterranean atmospheric circulation,
resulting in generally dry, hot summers and cold winters. Indeed, dew point temperatures at the station between 2018 and 2021
after sunset exceeded 15 °C only during summer (Fig. 2). The prevailing wind direction occurring at the site is W-WSW-SW
(Fig. 3).

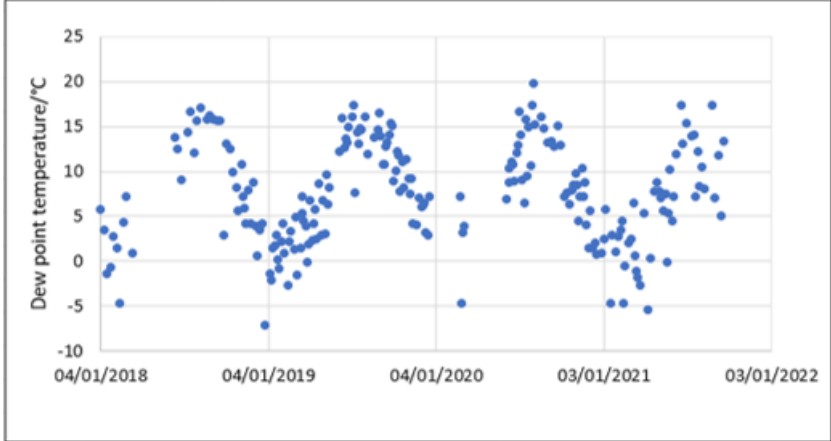

**Figure 2: Dew point temperature time series at CIAO in the timeframe 2018-2021 calculated from the RH and temperature values measured at the ground level with the sensors of VAISALA RS41 radiosondes, typically launched twice a week between 30 and 120 minutes after sunset.**

Most of the surrounding land is classified as arable crops in non-irrigated areas, followed by broad-leaved woods and coniferous forests, sclerophyllous or wooded/shrubby areas and natural grazing areas and grasslands (http://rsdi.regione.basilicata.it, last access: 28 November 2023).

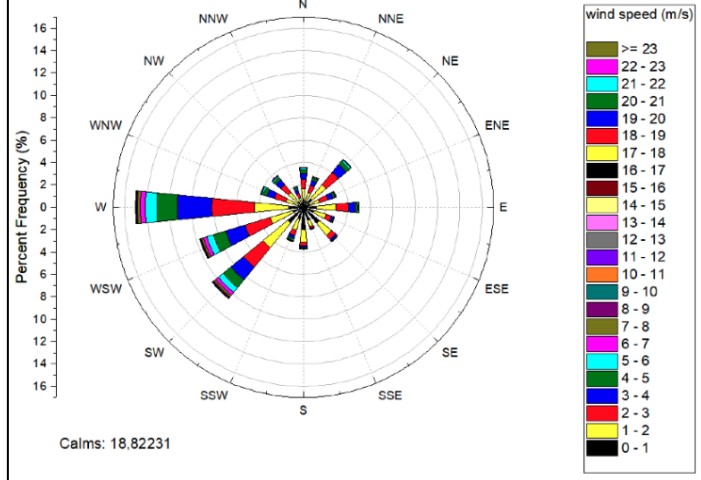

**Figure 3: Wind rose diagram at CIAO in 2004-2017 obtained from continuous measurements of the automatic weather station VAISALA MILOS520 with temporal resolution of 1 minute.**

CIAO's mission is to improve the knowledge of atmospheric processes and their role in meteorological phenomena, climate change and air quality. Given the coverage and global relevance of the processes studied, fundamental aspects of the activities



and approaches adopted are the development of internationally recognized Standard Operating Procedures, the open data policy
and the full sharing of methodologies and know-how.
CIAO provides free and open access to national and international users like researchers, Small and medium-sized enterprises
(SMEs), students and citizens. At the present time, CIAO extends its outreach through the ATMO-ACCESS Trans-National
Access program (https://www.atmo-access.eu/second-call-for-access/, last access: 2 December 2023). This program allows
participants to engage in research on aerosols and their effects, to learn techniques and methods, and to contribute to
instruments, or to collaborate with the team.
The research activities of CIAO revolve around the long-term observations of aerosols, clouds, trace gases and greenhouse
gases within the European research infrastructures ACTRIS and ICOS (Integrated Carbon Observing System), as well as
around the participation of CIAO in reference observational programs and networks on a global scale, such as GRUAN (GCOS
Reference Upper-Air Network) and GALION (GAW Aerosol Lidar Observation Network). The observational strategy is
organized to provide quality assured measurements for satellite validation and model evaluation and to fully exploit the synergy
and integration of the active and passive sensors for the improvement of the atmospheric characterization (e.g., Pappalardo et
al., 2004;; Mona et al., 2009; Boselli et al., 2012; Ilić et al., 2022). The complete list of CIAO publications is available at
https://ciao.imaa.cnr.it/publications/.
For what concerns aerosol measurements, CIAO due to its geographical position as well as the low aerosol background
concentration is interesting for studying particles of natural origin such as desert dust and volcanic ash clouds. The site is
regularly affected by Saharan dust intrusions (e.g., Mona et al., 2006; Mona et al., 2014; Binietoglou et al., 2015; Soupiona et
al., 2020) has been reached by volcanic aerosol at the level of free troposphere during the eruptions of Etna (e.g., Pappalardo
et al., 2004a, Villani et al., 2006) and Eyjafjallajökull (Madonna et al., 2010; Mona et al., 2012; Pappalardo et al., 2013 )
volcanos in 2002 and 2010, respectively, and stratospheric layers (e.g., Sawamura et al., 2012). In recent years, the observatory
has become actively involved in the study of smokes originated by wildfires occurring both at short-range, spreading with
increased frequency in the surrounding forestry areas during the summer period (De Rosa et al., 2022), and long-range
transported plumes, such as the autumn 2020 California wildfires whose smokes transported in the stratosphere reached the
site within 13 days (Baars et al., 2019).

## 3 Remote sensing measurements

Remote sensing measurements have been the backbone of the research activity at CIAO since its beginning in the early 2000s,
with the scientific goal of providing long-term measurements for the climatology of aerosol and cloud properties. The main
research lines currently include: a) development of advanced lidar systems for the study of aerosols and aerosol-cloud
interactions; b) design and implementation of new products (such as aerosol typing, aviation-specific products, and
atmospheric boundary layer height); c) development and implementation of open data and FAIR data management policies
within the ARES node of the ACTRIS Data Center and RIs in the environmental field; d) development and implementation of





access policies to European RIs; e) deep/machine learning and signal processing applied to Earth observation; f) studies
integrated with transport models, satellite data and climate models; g) harmonization of the time series of measurements of
atmospheric variables; h) measurement campaigns for validation and integration with satellite data; i) networking at European
and global level.
Besides the compliance to the ACTRIS guidelines, all the remote-sensing measurements performed at CIAO are designed to
be in line with the main ground-based observation networks (i.e., EARLINET, CloudNet, AERONET, GRUAN, GALION)
and the major international standards provided by the WMO/GAW 2016, aiming at establishing a long-term, harmonised and
statistically significant database of measurements for climatological studies (Matthias et al., 2004).
The active remote-sensing instruments operative at CIAO include multi-wavelength Raman and polarization lidars,
ceilometers, Doppler lidars and polarimetric Doppler radars, and the passive ones include microwave radiometers,
photometers, and a high-resolution Fourier-Transform Infrared (FTIR) spectrometer.
With respect to the status of CIAO reported in previous papers (e.g., Madonna et al., 2011), some instruments are still operating,
some have been replaced by more recent and advanced ones, and new instruments for increasing the observational capabilities
have been added. The complete list of the remote sensing suite is reported in     Table 1.

| Aerosol Remote Sensing | Cloud Remote Sensing | Trace gases Remote Sensing |
|---|---|---|
| Fixed multi-wavelength Raman lidar | Ka-band Doppler radar Metek MIRA-35 | FTIR Bruker 125HR |
| Mobile multi-wavelength Raman lidar | Compact Ka-band Doppler radar MIRA 35C | |
| Lidar and optical laboratories | W-band Doppler radar RPG-FMCW-94 | |
| | K-band Doppler radar Metek MRR-PRO | |
| MUSA Transportable Fixed multi-wavelength Raman lidar | Microwave radiometer RPG-HATPRO-G5 | |
| Scanning UV Raman lidar | Ceilometer Vaisala CL51 | |
| Automatic sun/sky/lunar photometer Cimel 318T | Ceilometer Vaisala CL31 | |
| | Ceilometer Lufft CHM15k | |
| | 2 Doppler lidars Halo Photonics Stream LineXR | |

**Table 1: List of the CIAO Remote Sensing instruments**

For the aerosol remote sensing, two new highly-advanced lidar systems have been recently installed at CIAO, one fixed and
one mobile. The first one will be an ACTRIS Observational Platform and the second one an Exploratory Platform available
even in combination with cloud remote sensing equipment. Both the lidars are capable of carrying out continuous
measurements going well beyond the required ACTRIS/EARLINET standards. They are able to provide measurements of
vertical profiles of several aerosol optical properties: backscatter coefficient and particle depolarization at 1064, 532 and 355
nm, and extinction coefficient at 532 and 355 nm, with the observational range starting from 200m up to at least 20km of
altitude. The fixed lidar is able to reach a measurement altitude range higher than 20km, being equipped with two telescopes,





and to provide vertical profiles of water vapor mixing ratio, this latter useful for investigating the impact of water vapour on
aerosol properties. On the other hand, the mobile system is more compact and transportable, and is used for field campaigns.
Both systems are part of the Centre for Aerosol Remote Sensing (CARS), one of the ACTRIS central facilities, that has the
mission to offer operation support to ACTRIS National Facilities operating aerosol remote sensing instrumentation. The two
systems are reference lidars for ACTRIS and offer services to test the performances of other lidar systems also through on-site
direct intercomparison campaigns using the mobile lidar.
Close by to the aerosol multiwavelength depolarization Raman, a triple mode photometer is operational within AERONET
(AErosol RObotic NETwork) and ACTRIS providing columnar aerosol optical depth measurements and columnar size
distribution information not only in daytime, but also in night-time under certain illumination conditions. CIAO is also
equipped with a lidar laboratory and an optical laboratory. The lidar laboratory is a facility that allows to implement and test
several and customized lidar configurations (fluorescence, HSRL, multiwavelength elastic/Raman/depolarization and water
vapour mixing ratio and liquid water, and rotational Raman for temperature) in a modular way. The optical laboratory allows
to test and characterize optical components and laser sources typically used in high power lidar systems. Both laboratories are
also part of CARS and are open to users who want to benefit from the offered services.
Besides the aerosol remote sensing instruments, cloud remote sensing equipment has also been updated and expanded with
additional complementary instruments, setting up both an Observational Platform and a mobile Exploratory Platform
compliant with ACTRIS/CloudNet requirements. Among the complementary instruments, Doppler lidars are capable of
measuring the profiles of horizontal and vertical wind and related fluid dynamic parameters through the troposphere. Finally,
a high resolution FTIR spectrometer has been added for performing remote sensing measurements of trace gases to complement
the other observations.
The availability of a large number of remote-sensing systems at the observatory has enabled the possibility to both compare
and combine different techniques for studying atmospheric parameters. The synergy between lidars and photometer
observations allows the retrieval of vertical profiles of aerosol concentration for total, fine and coarse components through
algorithms like GARRLiC (Generalised Aerosol Retrieval from Radiometer and Lidar Combined data; Lopatin et al., 2013)
and POLIPHON (Polarization Lidar Photometer Networking; Mamouri and Ansmann, 2016; 2017). Ceilometers have shown
good capabilities in the detection of aerosol plumes even if with some limitations (Wiegner et al., 2014; Madonna et al., 2015).
The combined use of ceilometers and multiwavelength polarization Raman lidars can be an added value for aerosol variability
investigation. Additional cloud information provided by 24-h ceilometers can be precious for cloud masking prior to the
analysis of lidar measurements for aerosol profiling. These aspects are currently under investigation and developments within
ACTRIS implementation.
The combination of lidars and radars also demonstrated the enhancing power of synergistic observations: combination of lidar
and radar measurements during the Iceland volcanic eruption in 2010 showed radar capability of detecting giant volcanic
particles (Madonna et al., 2010; Madonna et al., 2013).





Synergistic approach has been proposed for the study of thin liquid water clouds, combining multi-wavelength lidar and
Doppler radar measurements (Rosoldi et al., 2022). It has been shown that microwave radiometer can be used to calibrate
Raman lidar measurements for water vapour profiling and that the synergy between these instruments is an effective means
for atmospheric water vapour monitoring (Madonna et al., 2006, Mona et al., 2007).
However, despite its huge potential in atmospheric research, there are two major drawbacks associated with the remote sensing
observations: the inability to conduct aerosol measurements under skies with low clouds and precipitations, along with the
impossibility of characterising the particulate properties near the ground. Therefore, the recent implementation of the in-situ
facility described in the next section is fundamental to achieve a complete characterization of the aerosol at the ground level
where the aerosol particles directly affect ecosystems and human health. In addition, the in-situ measurements include the
valuable chemical characterization of the particulate matter (PM), thus providing a deeper comprehension of the aerosol type,
the source apportionment and the mixing atmospheric processes.

**4 Description of the aerosol in-situ facility**
**4.1 Overview**
The in-situ facility recently installed at CIAO comprises two main parts (Fig. 4): a field laboratory for aerosol online
measurements with continuous instrumentation and PM samplers and a chemical laboratory for the post-sampling analysis of
particulate collected over the filters.

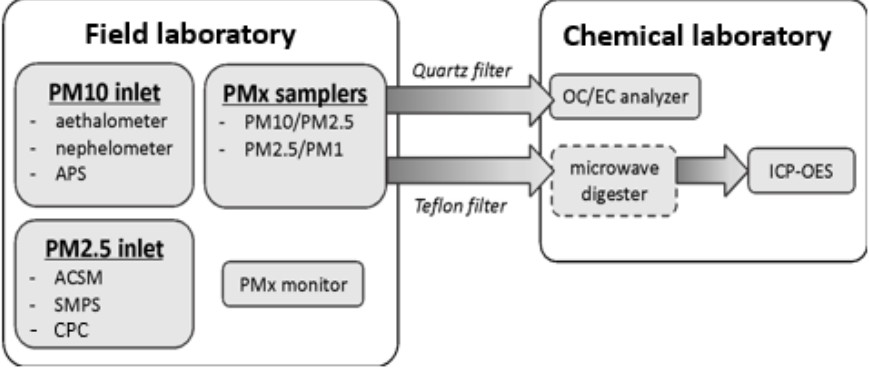

**Figure 4: Outline and workflow of the in-situ facility at CIAO.**

The shelter has been designed according to the ACTRIS guidelines and recommendations (Wiedensohler et al., 2014), with
the instrumentation arranged as follows: a dual spot aethalometer (AE33, Magee Scientific), a multi-wavelength integrating





nephelometer (AURORA 3000, Ecotech) and an aerodynamic particle sizer (APS 3321, TSI) located under a common $PM_{10}$
(aerosol particles with an aerodynamic diameter less than 10 μm) inlet; a time-of-flight aerosol chemical speciation monitor
(ToF-ACSM, Aerodyne Research), a scanning mobility particle sizer (SMPS, 3938-TSI) and a condensation particle counter
(CPC, 3750-TSI) placed under a $PM_{2.5}$ (aerosol particles with an aerodynamic diameter less 2.5 μm) common inlet.
Additionally, two PMx samplers (SWAM 5a-Dual Channel Monitors, FAI Instruments) and a PMx monitor (EDM 180,
Grimm) are placed as standalone instruments with individual sampling lines.
Particular attention has been devoted to the design of the common inlets and the sampling lines. The $PM_{10}$ and $PM_{2.5}$ common
impactor type inlets, operating at a flow rate of 16.7 l $min^{-1}$, are compliant with EN 12341 and EN 14907 standards,
respectively. The main challenge when transporting aerosol to collectors and aerosol measuring instrumentation is to avoid
aerosol losses. Therefore, firstly, the internal diameter of the main sampling pipe of the common $PM_{10}$ and $PM_{2.5}$ inlets must
be such as to ensure that the sampled air has a laminar flow along the entire path (Reynolds number less than 2000) to minimise
the loss of particles by diffusion and inertia. The instrument sublines (characterised by smaller inside diameters) are connected
to the common inlet through an isokinetic flow splitter where the sample flow velocity is almost equal to the velocity of the
main flow. Moreover, the ends of the tube in the isokinetic flow splitter must be sharp to ensure a homogeneous distribution
of the air sample to the instruments. Another key feature of the splitter is that the sample is collected from the core of the main
aerosol flow rather than from streamlines near the wall of the main pipe, therefore, ensuring a representative sampling
(especially for coarse and nanoparticles).
The technical details of the common inlets and isokinetic splitters are shown in Table 2.

| Common Inlet | | | | Isokinetic splitter | | | | |
|---|---|---|---|---|---|---|---|---|
| Inlet | Flow rate (l $min^{-1}$) | Int. Diameter (mm) | Speed (m $s^{-1}$) | Reynolds Number | Instrument | Int. Diameter (mm) | Flow rate (l $min^{-1}$) | Reynolds Number | Speed (m $s^{-1}$) |
| PM 10 | 16.7 | 21.2 | 0.8 | 1135 | Aethalometer | 8 | 3 | 885 | 1.6 |
| | | | | | Nephelometer | 8 | 5 | 885 | 1.6 |
| | | | | | APS | 4.4 | 1 | 320 | 1.09 |
| | 16.7 | 21.2 | 0.8 | 1135 | SMPS | 4.4 | 2 | 655 | 2.2 |
| | | | | | CPC | | | | |





| PM 2.5 | | | | | TOF-ACSM | 8 | 3 | 530 | 1 |
|---|---|---|---|---|---|---|---|---|---|

**Table 2: Technical details of the common inlets and isokinetic splitters.**
All the sampling tubes are kept as short as possible and are placed in vertical position with bends and connectors avoided as
much as possible to suppress potential sources of turbulence, which would result in additional losses of particles. In addition,
the tubes are made of polyurethane antistatic material to guarantee perfect dissipation of accumulated static electricity, because
the static charges may remove significant portions of the aerosol to be sampled. The inlets on the rooftop of the field laboratory
are placed at one metre from each other and height of 1.5-2.0 m above the roof with the aim of minimising local influences
and potential interferences in the sampling process.
In compliance with the ACTRIS indications, all the instruments in the laboratory are equipped with a Nafion dryer tube which
keeps the RH well below 40%; under this threshold, in fact, changes in particle diameter due to RH variations are expected to
be lower than 5%, thus obtaining comparable data, independent of the hygroscopic behaviour of the aerosol particles.
Moreover, the upstream drying prevents the possible instrument damage caused by water condensation.
The Nafion dryers of Aethalometer, Nephelometer, APS and ACSM operate in a reflux mode, shown in Fig. 5, which returns
the dry sample back to the dryer for use as the purge after it has gone through the analyzer. Since this method uses all the dry
sample as purge air, only the sample flow required for analysis passes through the dryer. This results in a high drying efficiency.
The vacuum on the purge air should be at least 15 inches Hg, with a higher vacuum preferable. This vacuum level is required
to provide the desired 2:1 purge to sample flow ratio based on the actual volumetric flow.

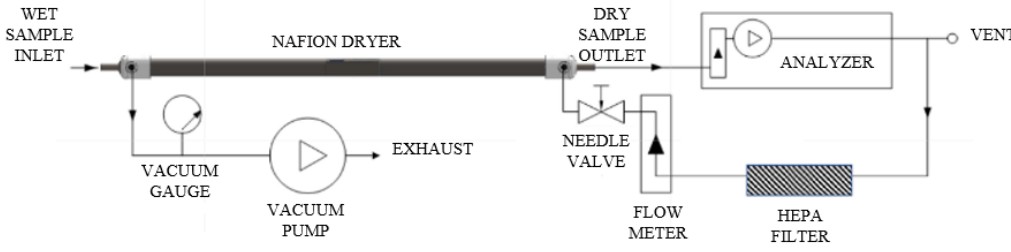


**Figure 5: Schematic Diagram - Nafion Reflux mode (MD-700-User-Manual, https://www.permapure.com).**
Instead, the Nafion dryer connected to the SMPS and CPC, since the instruments need n-butanol as a working liquid for the
growth of aerosol particles, cannot operate in reflux mode but operates in counter flows using air dry coming from a compressor
(Acoem 8301 LC-H Zero Air Generator) (Figure 6).



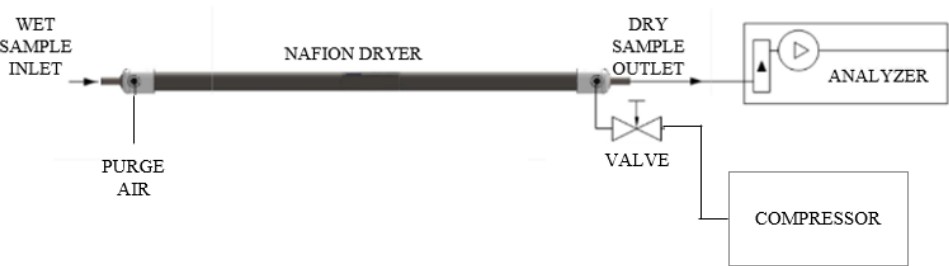


**Figure 6: Schematic Diagram - Nafion counter flows mode**.


Moreover, at the input of each instrument, there is a high-resolution sensor connected to a software for continuous monitoring
(every minute) of relative humidity and temperature. The accuracy of the sensors is 2.5% for the RH and 0.5°C for the
temperature.
Lastly, in order to limit the temperature variation around the instruments, a continuously operating air conditioning system set
at 23 °C has been installed in the laboratory.
As previously mentioned, the in-situ facility is complemented by the chemical laboratory which enables complementary
measurements on the particulate-loaded filters coming from the PMx samplers, that is not possible to obtain with the continuous
instrumentation. The chemical laboratory include: an inductively coupled plasma optical emission spectrophotometer (ICP-
OES, series 5800, Agilent) used to perform the analysis of trace metals, and a multi-wavelength OC/EC analyzer (DRI model
2015, Magee Scientific) used to analyse the carbonaceous fraction of the collected particulate.
**4.2 Instrumentation under the common PM$_{10}$ inlet**
As reported above, a PM$_{10}$ common inlet is used to feed the aethalometer, the nephelometer and the APS. The aethalometer is
a key instrument for wildfire and pollution characterization, being able of detecting the fraction of particulate which absorbs
light, known as Black Carbon (BC), formed during the incomplete combustion of carbonaceous matter from biomass burning
and fossil fuel (Petzold et al., 2013). According to the ACTRIS guidelines, the AE33 aethalometer operating at seven different
wavelengths in the range 370-950 nm is used for the real-time monitoring of the concentration of BC. Briefly, the principle of
the aethalometer is to measure at given time intervals the attenuation of a light beam (at 880 nm) transmitted through a filter
where the particulate is continuously collected; the rate of change of optical transmission combined with the air flow rate
monitored through a mass flowmeter permits to determine the absorption coefficient, then converted into BC concentration by
means of the mass-absorption cross section. The dual spot technology refers to the contextual measurement of transmitted light
intensities through two separate spots of the filter at different loading levels, thus allowing to compensate for the so-called
loading effect largely described by Drinovec et al. (2015). The aethalometer is equipped with a sample stream dryer (Magee
Scientific) exploiting a semi-permeable Nafion membrane which keeps the RH well below 40%.



Among the other in-situ instruments placed under the $PM_{10}$ inlet, the nephelometer can be considered in a certain way
complementary to a ground-based lidar, expecting therefore to provide optical parameters consistent with those obtained from
the lidar within the atmospheric planetary boundary layer (PBL). However, when the the PBL is particularly shallow (e.g.,
during wintertime), the nephelometer becomes the only tool to obtain the optical parameters of the aerosols residing within the
first hundreds of metres from the ground. The ACTRIS-compliant integrating nephelometer AURORA 3000 is used to measure
the total scattering ($\sigma_{sp}$) and the backscattering ($\sigma_{bsp}$) coefficients (integrating within the angular range 9°-170° and 90°-170°,
respectively), both correlated to the particle concentration (i.e., extensive properties). The peculiarity of the instrument is the
utilisation of a light source emitting at three distinct wavelengths: the light at 635 nm (red) interacts strongly with large
particulate matter such as desert dust and sea salt; the light at 525 nm (green) interacts strongly throughout the human range
of visibility (smog, fog, haze); the light at 450 nm (blue) interacts strongly with fine and ultrafine particulates, such as wood
fires and automobile combustion particulate. The nephelometer is equipped with a 36-inch-long Perma Pure Nafion MD-
700 in order to prevent condensation of water droplets over the particles, which would increase their size and significantly
change their scattering characteristics.
Lastly, the APS spectrometer provides high-resolution real-time aerodynamic measurements for the coarse fraction of the
particulate (Peters et al., 2003). The optical size range of the APS is from 0.37 to 20 µm, but since the spectrometer is connected
to a $PM_{10}$ inlet and the counting efficiency of APS below 0.8 µm aerodynamic diameter rapidly decreases and is unstable, the
realistically size range is from 0.8 to 10 µm. The APS is based on the time-of-flight particle sizing, in which the aerodynamic
size of a particle determines its rate of acceleration, with larger particles accelerating more slowly due to increased inertia; the
time of flight between two laser beams is recorded and converted to aerodynamic diameter using a calibration curve. The
instrument measures in parallel the light scattering intensity of the sized particles in the equivalent optical size range from 0.8
to 10 µm, thus providing further insights into the aerosol nature and composition.
The APS is connected to the sampling line just with the inner nozzle (sampling $1\,l\,min^{-1}$) from the common sampling line and
the flow is dried by a 12-inch Perma Pure Nafion, while taking the additional sheath flow ($4\,l\,min^{-1}$) from the air compressor.
**4.3 Instrumentation under the common $PM_{2.5}$ inlet**
Even though the general ACTRIS recommendations for the in-situ measurements involve the analysis of the $PM_{10}$ fraction,
the CPC, the SMPS and the ACSM represent an exception and are more conveniently placed under the cut-off size of a $PM_{2.5}$
inlet. The ACTRIS-compliant CPC is used to measure the number concentration of aerosol particles with diameter > 10 nm.
In the CPC, an aerosol sample is continuously drawn through a heated saturator where the butanol is vaporized and diffused
into the sample stream. Together, the aerosol sample and *n*-butanol vapour pass into a cooled condenser where the *n*-butanol
vapour becomes supersaturated and condenses on the particle surface causing them to grow. The particles are then counted
individually as they pass through a laser-based optical detector.
Regarding the SMPS, it is an instrument of interest for CIAO, being able to provide the size distribution and concentration of
the fine fraction of the particulate in the size range 10 nm – 800 nm. It consists of four components in sequence: 1) a pre-





impactor which removes particles larger than the fixed upper limit of size; 2) a bipolar diffusion charger (model 3082, TSI)
which confers a characteristic stationary charge distribution to the polydisperse particles by using a radioactive source (Kr-
85); 3) a differential mobility analyzer column (DMA, model 3083, TSI) which separates the particles according to their
electrical mobility by varying continuously the applied voltage within the column (Schmid et al., 2007); and 4) a condensation
particle counter (CPC, model 3750, TSI) where the classified monodisperse particles are counted after condensation of *n*-
butanol on their surface.
The CPC and the SMPS are connected to the same 24-inch Perma Pure Nafion via a T-flow splitter in order to keep the RH
below 40%. Moreover, a dry sheath air is needed for the SMPS to ensure particle sizing inside the DMA with a minimum
fluctuation in RH and for this purpose a Silica Dryer Tube (model 3082, TSI) is incorporated in the DMA sheath flow system,
which is a closed loop.
For what concerns the aerosol mass spectrometry techniques, the     ToF-ACSM (Aerodyne Research) has been shown to be
perfectly suited for the ACTRIS observatory platforms, having been designed to provide continuous and unattended
measurements for aerosol monitoring on the timescale of years. The chemical speciation with high temporal resolution is a
unique feature of the ACSM technology, unobtainable with conventional filter sampling and subsequent post-processing
chemical methods; moreover, the ACSM is not subjected to sampling artefacts that affect the collection of semi-volatile PM
components by means of filters (Viana et al., 2006; Kim et al., 2015). The ToF-ACSM chosen for CIAO was introduced in
2013 (Fröhlich et al., 2013), providing a higher mass resolution (i.e., $m/\Delta m = 600$) and superior detection limits (i.e., $< ng\ m^-$
$^3$) with respect to the previously developed quadrupole-ACSM (Ng et al., 2011) for a time resolution of 30 min. The instrument
measures the mass and chemical composition of non-refractory submicron aerosol particles – i.e., organic substances, nitrates,
sulphates, ammonium, and chloride – thus generating an invaluable database for the research community to characterise the
particulate sources and evolution. The operational principle of the instrument is briefly described in the following: the aerosol
enters the inlet where the aerodynamic lens efficiently samples and focuses submicron particles to the subsequent vacuum
chamber; here, the particles impact on a resistively heated porous tungsten surface at approximately 600 °C which vaporises
the non-refractory particulate; the vaporised matter is subsequently ionised by electronic impact and detected through the ToF
analyzer. In this case, the 24-inch Nafion dryer installed upstream the instrument eliminates the complicating inlet effects due
to particle composition dependent water uptake (Middlebrook et al., 2012).
ACSM was installed in February 2023 and worked for some months in almost continuous way. Then some interventions were
requested to accomplishing the optimization requests from ACTRIS aerosol in situ central facility, and the ACSM restarted
operations just recently in April 2024. Anyhow the 3 months of almost continuous measurements performed in 2023 already
provide some insights about aerosol present at the surface in Potenza. Figure 7 reports daily concentrations for the 4
components as measured by ACMS in February-March-April 2023 period. Median values are preferred to mean ones for
avoiding the strong influence of outliers and spikes in the reported values. Monthly pie charts show the relevance of the
difference components for each one of the 3 months. As a general comment, we could say that the Potenza site is clearly a
rural site with low PM concentration and a very high contribution of the organic substances (see for comparison as example



Atabakhsh et al., 2023 and Zhao et al., 2020). The observed increasing in the total concentration but more pronounced in the
organic component could be related to tree pollen events typically occurring in such period.

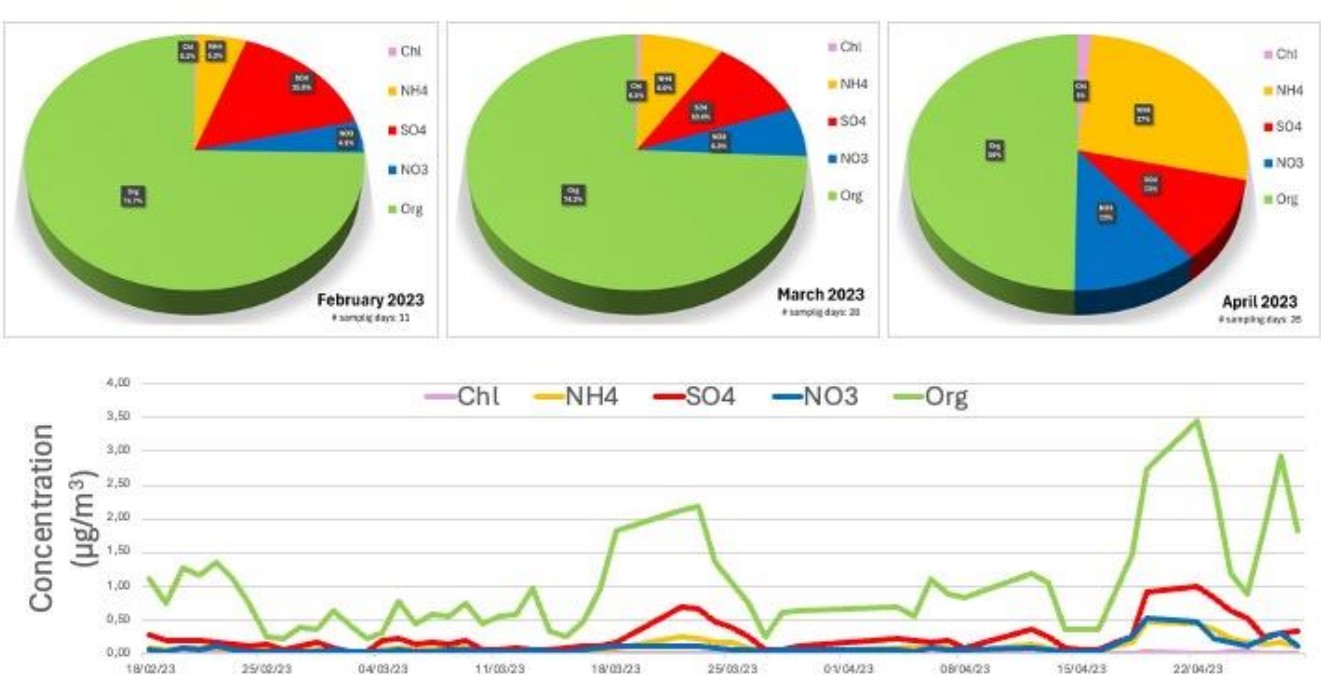


**Figure 7: Daily medians of the mass concentration and mass fraction of each of the 4 chemical components of non-refractory**
**submicron aerosol particles observed at CIAO in February – March – April 2023.**
**4.4 PMx samplers and PMx monitor**
Additionally to the online instruments report above, the field laboratory is equipped with two PMx samplers for the continuous
sampling and concentration measurement of $PM_{10}$, $PM_{2.5}$ and $PM_1$ (aerosol particles with an aerodynamic diameter less than 1
μm) mass fractions collected over both Teflon and quartz filters; the determination of the mass of collected samples is based
on the $\beta$-ray attenuation equivalent method, which strongly reduces the workload and the operator-associated variability if
compared to the standard gravimetric method (Baltensperger et al., 2001). In particular, the device measures the attenuation
of $\beta$-ray across the filter medium which collects particulate matter, and the attenuation of intensity in β-ray is proportional to
the amount of material present. Each PMx sampler is equipped with two independent sampling lines (i.e., $PM_{10}$/$PM_{2.5}$ and
$PM_{2.5}$/$PM_1$), thus enabling the simultaneous collection of different PM fractions on independent filters. According to the
workflow reported in Fig. 4, the particulate collected over the filters is subjected to further analysis within the chemical
laboratory: the $PM_{10}$, $PM_{2.5}$ and $PM_1$ collected over 24h on Teflon filters are analysed to determine the concentration of metals





356 by means of the ICP-OES. On the other hand, the $PM_{2.5}$ collected over 24h on quartz fibre filters are analysed to quantify the

357 organic carbon (OC) and elemental carbon (EC) fractions using the thermal optical method by the OC/EC analyzer; the

358 utilisation of quartz fibre filters for the OC/EC analysis is strictly recommended by the WMO/GAW 2016 guidelines, and it

359 constitutes the only exception to the Teflon filters commonly used for other analyses. In fact, the particulate collected on Teflon

360 filters is not limited to ICP-OES analysis but can also be analyzed through alternative techniques such as X-ray fluorescence

361 (XRF) and Particle Induced X-ray Emission (PIXE) in order to find complementarities between the three techniques for the

362 determination of a range of metals.

363 Furthermore, even if not included in the mandatory ACTRIS variables to be measured, the mass concentration for the cut-off

364 diameters of $PM_{10}$, $PM_{2.5}$ and $PM_1$ belongs to the set of standard measurements to monitor the particulate matter, providing

365 insight into the separation of fine and coarse particles within the aerosol.

366 The PMx monitor operating at CIAO currently represents one of the main automated measurement systems for studying the

367 concentration levels of particulate matter in ambient air. Based on the detection principle of the light scattering at the level of

368 single particles, the system offers simultaneous real-time measurements of $PM_{10}$, $PM_{2.5}$ and $PM_1$ and particle number

369 distribution with a resolution of 0.1 µg m$^{-3}$.

### 4.5 Chemical laboratory

371 The CIAO chemical laboratory is equipped with an ICP-OES and an OC/EC analyzer. The ICP-OES (5800 series, Agilent) is

372 used to determine the qualitative and quantitative elemental composition of the metals present in the atmospheric particulate

373 with high sensitivity, at values below the 1 µg l$^{-1}$ limit for certain elements. The metals are introduced into the atmosphere

374 from various anthropogenic and natural sources. Anthropogenic metals are released into the atmosphere during combustion of

375 fossil fuels and wood, as well as during high temperature industrial processes and waste incineration; natural emissions result

376 from a variety of processes acting on crustal minerals, including volcanism, erosion, surface winds, forest fires and ocean

377 evaporation (Allen et al., 2001; Pakkanen et al., 2001; Rajšić et al., 2008). Various metals are used as marker for the

378 identification of emission sources: aluminium and silicon are primarily derived from soil and rocks (crustal elements); sodium

379 and chlorine are typically associated to marine aerosols; arsenic, cadmium, manganese and lead mostly derive from combustion

380 of fossil fuels occurring at high temperature, to name a few. The ICP-OES analysis of particulate matter requires a preliminary

381 microwave digestion of the filter in acidic conditions to extract the metals, carried out by means of a microwave digester

382 (ETHOS UP, FKV). The obtained liquid sample is then nebulized and introduced into the plasma as an aerosol suspended in

383 the argon gas: due to the high temperatures within the plasma (7000 – 10000 K), a significant fraction of most elements exists

384 as atoms or ions in the excited state, causing an intense polychromatic emission which continuously brings back the elements

385 to their ground state. The polychromatic emitted light is dispersed into individual wavelengths by a polychromator and detected

386 by a photosensitive charge-coupled device (CCD). The concentration of each metal in the sample is obtained by using a

387 calibration curve referred to a solution containing the analysed elements of known concentration.


The multi-wavelength OC/EC analyzer (2015 DRI, Magee Scientific) compliant with ACTRIS is used to quantify the total
carbonaceous content of the particulate matter (total carbon, TC) and the OC and EC subfractions. EC is essentially a primary
pollutant, emitted directly from the incomplete combustion of fossil fuels and the pyrolysis of biological material during
combustion, whereas OC can be directly emitted from the incomplete combustion of organic materials and the degradation of
carbon containing products such as vegetation – primary OC – or produced from atmospheric reactions, involving gaseous
organic precursors, i.e., secondary OC (Zhou et al., 2006). The operational principle of the thermal/optical analysis is based
on the preferential desorption of OC and EC materials under different temperatures and atmospheres programmed within
specific thermal protocol, such as the EUSAAR_2 (Cavalli et al., 2010) which is currently used within the ACTRIS community.
OC usually desorbs under a non-oxidising helium atmosphere at temperatures up to 570 °C, while the EC is combusted in an
oxidising atmosphere with 2% $O_2$ at temperatures up to 850 °C. However, since part of the OC turns into the light-absorbing
pyrolytic carbon which desorbs during the oxidising mode, the correct discrimination between the OC and the EC fractions is
conveniently identified with the point at which the light transmission reaches the pre-pyrolysis value. The liberated carbon is
then completely oxidised to carbon dioxide passing through a heated catalyst $MnO_2$ and finally quantified by an NDIR detector.

**5 Synergistic deployment of aerosol remote sensing and in-situ measurements**

Synergistic approaches combining aerosol profiling and in-situ measurements are one of the most beneficial strategies in
aerosol research, allowing an accurate typing and estimation of the impacts of particulate matter (Molero et al., 2020). Remote
sensing techniques provide the vertical profile of the particle size distribution of the aerosol as well as further physical and
optical properties useful for understanding complex atmospheric phenomena (Vratolis et al., 2020); however, they are not able
to provide information under cloudy sky conditions or at the ground level, where the identification of aerosol type is only
possible using the in-situ instrumentation. The in-depth typing of the aerosols requires the information on the chemical
composition, attainable only by means of in-situ measurements. The complete set of data resulting from the combined
approaches is crucial for identifying the sources and the evolution of concentration levels of particulate matter over time (Bressi
et al., 2021), and it is of paramount importance for the implementation of controls or policies to reduce aerosols that negatively
affect air quality and public health.
The complete picture of the aerosol-typing is also expected to clarify further the climate effects of particulate matter. In fact,
the estimation of the radiative effect of atmospheric aerosol requires the knowledge of multiple parameters, including the
aerosol load, the optical properties, the chemical composition, the presence of clouds and the albedo of the underlying surface.
The accurate identification of aerosol types is also needed to improve the understanding of atmospheric dynamics and long-
range transport, to improve satellite aerosol retrieval algorithms, and to validate climate models.

The multiwavelength polarisation Raman lidar is a well-established active remote sensing technique for the detection and
characterization of aerosol-types (Nicolae et al., 2018; Papagiannopoulos et al., 2018). Specifically, it can provide vertically





resolved information on extensive (e.g., aerosol backscatter coefficient, aerosol extinction coefficient and volume
depolarization ratio) and intensive (e.g., Ångström exponent, lidar ratio and particle depolarization ratio) aerosol optical
properties. The extensive properties depend on the aerosol concentration, whilst intensive ones are type-sensitive providing
indication about the particle size, shape, and indices of refraction that allow for the characterization of different aerosol types.
Nevertheless, the intensive properties might not be sufficient to guarantee accurate typing, as some aerosol types (e.g., volcanic
and desert dust particles) have very similar intensive properties but are attributed to different sources and generating
mechanisms. For this reason, the discrimination of aerosol particles that typically have the same optical characteristics calls
for the combined use of lidar observations and transport model simulations.
Finally, the aerosol in-situ observations can help in the assessment of the uncertainty of remote sensing-retrieved products like
mass concentration, refractive index and fine-particle concentration obtained through inversion algorithms (e.g., Veselovskii
et al., 2012; Lopatin et al., 2013).

In the following subsections we present three emblematic cases recurring at CIAO where the combined deployment of the in-
situ and remote sensing observations is expected to be of added value: 1) Wildfires become more and more relevant in the
Mediterranean, especially in view of the changing climate that is expected to increase temperature and in turn will affect their
frequency, duration and intensity in the next decades. In this context, small and local fires are widely distributed and their
characteristics and assessment could be important at global level. De Rosa et al. (2022) showed with the use of lidar
observations that fresh fires can be surprisingly characterised by low absorption; this would imply a different impact of local
fires in the radiation budget which requires investigation and validation by means of in-situ measurements. 2) Local pollution
during winter and adverse weather can be investigated in a more exhaustive manner only by in-situ observations, since lidar
observations provide very little information due to the generally low and unresolved by lidars PBL height. 3) Desert dust
intrusions often reach Europe and especially the Mediterranean Basin affecting local air quality, health and ecosystem and
socio-economic sectors (e.g., Monteiro et al., 2022). Given all the above, the deployment of in-situ measurements at well-
equipped sites like CIAO is crucial to quantify the impact at the ground level.
**5.1 Local wildfires**
The study of smokes from wildfires spreading in short distance represents a great example for a synergistic approach based on
remote-sensing and in-situ techniques. In such a case, the smoke particles spread mainly at low levels and deposit fast on the
ground, where in-situ measurements are the only tool to provide reliable information to support and integrate what is observed
above medium overlap region, a prerogative of remote-sensing techniques.
The multiwavelength polarisation Raman lidar is a well-known tool to study smoke layers in the atmosphere, being able to
separate aerosols according to their specific optical signature (Ohneiser et al., 2021). Specifically, a sign of the dominance of
smoke in the aerosol layer is the aerosol extinction-to-backscatter ratio (the so-called lidar ratio, S) at 532 and 355 nm, which
is typically high (i.e., > 50 sr) as a consequence of the presence of absorbing BC produced during the biomass burning;





moreover, the ratio of S measured at the different wavelengths may be used as an indicator of the phase of the ongoing wildfire
(e.g., Nicolae et al., 2013). Other lidar parameters largely used to investigate the smoke are the particle linear depolarization
ratio (PLDR) and the Ångström exponent (AE), which provide information about the shape and the size of the particles,
respectively. In the case of a local wildfire, the observation of quasi-spherical and relatively small particles is expected, since
the newly produced smoke particles do not have the time to undergo modifications during transport.
The Ångström absorption and scattering exponents (AAE and SAE) - derived from the aethalometer and nephelometer
measurements, respectively - provide the optical typing of the smoke, with the value of AAE expected to correlate with the
lidar observations (Cazorla et al., 2013) and, therefore, to the nature of spreading fire.
Among the aerosol in-situ instruments, the aethalometer is crucial to study smokes produced during wildfires, being able to
quantify the BC originated from the incomplete combustion of carbonaceous matter and providing an estimate of the biomass
burning (BB) apportionment to the overall BC (Sandradewi et al., 2008). Furthermore, particles resulting from the incomplete
combustion have been reported to contain a significant organic carbon fraction, including numerous known toxic and
carcinogenic polycyclic aromatic hydrocarbons (PAHs) (Nelson et al., 2021). To access relative amounts of organic carbon,
the method involves comparing two optical indicators of carbonaceous particulate matter derived from the aethalometer
measurements: BC at 880 nm, measuring elemental carbon that absorbs a broad spectrum of wavelengths, and UVPM at 370
nm, measuring particulate matter that, due to increased organic carbon content, absorbs disproportionately in the UV range
compared to BC (Olson et al., 2015), UVPM, also known as brown carbon, is associated with toxic species such as PAHs and
has been observed to be elevated in smoke resulting from biomass burning (Huangh et al., 2018). Additionally, the OC/EC
thermal/optical analysis on $PM_{2.5}$ fraction is very important because the increase of organic carbon and elemental carbon
concentrations has been the most indicated as an element that reflects wildfire emissions. The fine particles, particles generally
2.5 µm in diameter or smaller, represent a main pollutant emitted from wildfire smoke so other important in-situ analyses of
the travelled smoke are the size distribution and concentration of fine particles by the SMPS and CPC, respectively. In fact, it
is expected that the fine mode will be more densely populated and concentrated during these events compared to the rest of
the year. Further confirmation of the increase in fine and ultrafine particulate matter during fire events is given by the
$PM_{2.5}/PM_{10}$ and $PM_1/PM_{2.5}$ ratios obtained from the real-time measurement of $PM_{10}$, $PM_{2.5}$ and $PM_1$ concentrations using the
PMx monitor. In fact, the mean fraction of fine PM ($PM_{2.5}/PM_{10}$) and ultrafine PM ($PM_1/PM_{2.5}$) is expected to be significantly
higher during the fire period compared to the non-fire period. Finally, the in-situ investigation of wildfire smoke is completed
by the chemical analysis obtained with the ToF-ACSM: in particular, key tracers of biomass burning organic aerosol in mass
spectra are the enhanced signals at m/z 60 and 73 Th attributed to $C_2H_4O_2^+$ and $C_3H_5O_2^+$ ions, respectively, coming from the
fragmentation of the so-called "levoglucosan-like" species originated from the pyrolysis of cellulose (Cubison et al., 2011).
Finally, the chemical analysis of the filters through the ICP-OES is fundamental for tracking the levels of potentially toxic
elements (PTEs) such as As, Sb, Cd, Hg, Pb, Cr, Cu, Ni, Se, Tl, Sn, V, and Zn. This monitoring is vital as these elements have
the potential to be released into the environment during wildfires, posing a threat to humans and animals when their absorbed
doses surpass the established reference values (Pacifico et al., 2023).





## 5.2 Local pollution in wintertime


Winter months commonly exhibit heightened air pollution levels, primarily attributed to temperature inversions. Inversion
occurrences involve a layer of warm air confining colder air and pollutants close to the ground, impeding their dispersion into
the atmosphere. Unlike summer air pollution, winter conditions result in the prolonged presence of pollutants, increasing the
likelihood of higher inhalation rates. This extended exposure raises health concerns for individuals, as reduced ventilation and
dispersion contribute to potential health effects.
Air quality near the ground during winter is expected to be dominated by local residential heating emissions with the
contribution of vehicle engine exhausts. For this season, the in-situ measurements represent the most viable way to investigate
the aerosol distribution and composition, while the deployment of remote sensing instruments (e.g., lidar) is limited by
instrumental and environmental factors. During wintertime, the condensation of water droplets (especially during nighttime)
along with the recurrent formation of cloud layers attenuate the laser beam, thus impeding the lidar/ceilometer measurements;
moreover, even under clear sky conditions, the particulate is usually confined within the first 300 m from the ground (i.e., the
typical PBL layer thickness in wintertime), where the active remote sensing techniques are not able to provide reliable results.
On the other hand, the in-situ instrumentations enable the analysis of the particulate matter collected at the ground level where
the pollutants highly concentrate as a result of the stagnant, dense and cold air. Among the in-situ measurements, aethalometer
plays a key role: the BC content of particulate matter originates mostly from the incomplete combustion of both fossil fuel and
biomass used as combustibles for domestic heating. The BC content is expected to be higher with respect to the background
summer levels, especially due to the contribution of local residential heating and the air stagnation. The BC source
apportionment is expected to be determined by both biomass burning fraction due to the residential wood burning and the
fossil fuel due to the traffic exhaust and residential heating.
The OC/EC analysis on the $PM_{2.5}$ fraction is expected to provide additional data to both support and integrate the results
obtained with the aethalometer (Schmidl et al., 2008; Gonçalves et al., 2010; Pio et al., 2011; Sirignano et al., 2019). The
nephelometer is expected to provide the optical parameters at the ground complementing the ones obtained by lidar
measurements in the 300 m-UTLS region; the total scattering coefficient $\sigma_{sp}$ and the backscattering coefficient $\sigma_{bsp}$ are related
to the concentration of particles, with a dominant response expected at 450 nm and relatively high values for the SAE,
corresponding to the fine and ultra-fine particles typically produced by heating emissions (Esteve et al., 2012). Further
elucidations on the nature and the origin of the particulate can be certainly obtained with the ToF-ACSM. In this case, an
accurate prediction of the chemical composition of the particulate is not a trivial task since many factors contribute to the
chemistry of the particulate and, as of today, there are no previous data reports for such type of analysis at the site in wintertime.
In principle, however, the chemical speciation of the $PM_1$ fraction from the ACSM is expected to put in evidence a prevalence
of the organic matter derived from the combustion processes. Moreover, as previously reported (Chen et al., 2012), during
wintertime the recurrent exceedances of the fine particle fractions may be due to the abundance of the secondary ammonium
nitrate ($NH_4NO_3$), attributed to residential wood combustion and diesel engines through the emission of nitrogen oxides (NOx)





from these sources. Finally, the importance of the chemical analysis of the filters must be underlined; through ICP-OES the
main metals present in the particulate can be analyzed (Na, Mg, Al, Ca, V, Cr, Fe, Mn, Ni, Cu, Zn, As, Mo, Sb, Cd, Ba, Pb)
which come from specific sources, such as the combustion of fossil fuels in industries or power plants or in vehicle combustion
engines, coal and wood combustion processes, non-combustion related emissions from vehicular traffic and dust resuspension
phenomena resulting from traffic (Dušan et al., 2017; Zhi et al., 2021).
In the following we investigate the average daily concentration of equivalent black carbon (eBC) obtained by the aethalometer
(Figure 8a), covering the period from June 2023 to April 2024, to have a first insights into air quality near the ground during
winter. Our analysis reveals no significant increase in eBC concentration during the winter months compared to background
levels observed in summer. However, when examining the daily average percentage of black carbon (BC) originating from
biomass burning (BB%), as determined by the Sandradewi model, in conjunction with the daily average temperature data
obtained from the Vaisala AWS310 weather station situated at the site (Figure 8b), an intriguing trend emerges. It becomes
evident that BB% is substantially higher during winter months than during summer months. Given the minimal occurrence of
wildfires and prescribed burns at the site during winter, the primary source of biomass burning influence can be attributed to
residential burning, a consequence of the notably low temperatures experienced during that period. These first data indicate
that the main source of BC during winter at our site is predominantly from local residential heating emissions.

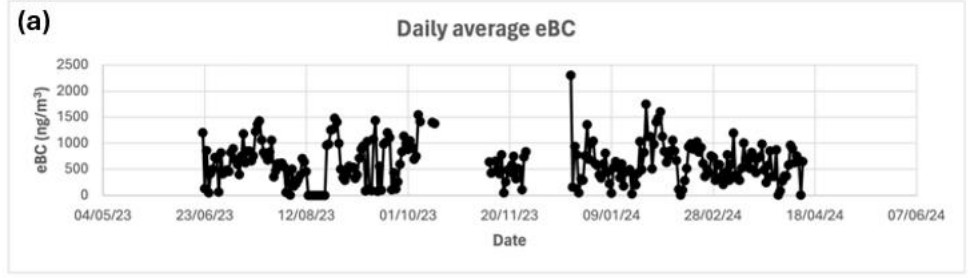

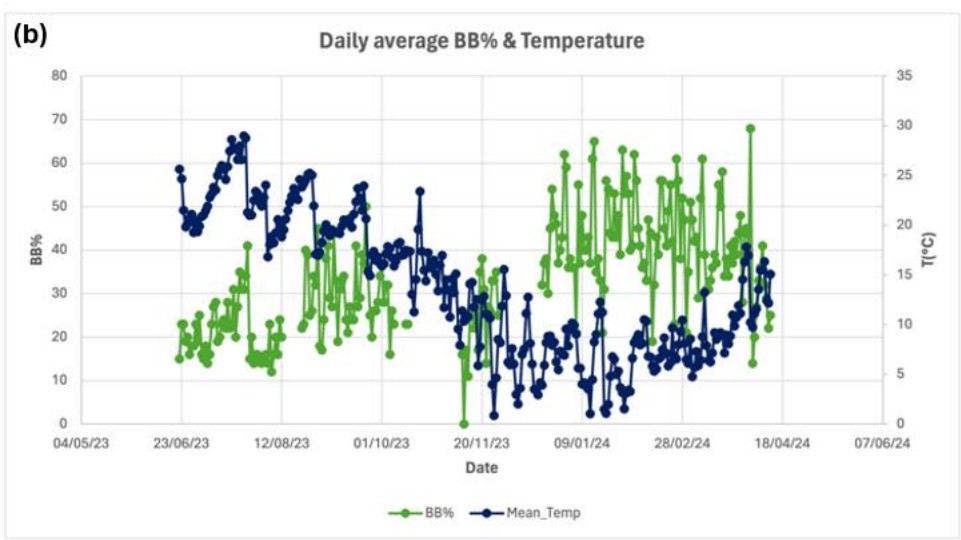






**Figure 8: Daily average eBC concentration obtained by aethalometer from June 2023 to April 2024 a), top panel daily average BB%
determined by the Sandradewi model from June 2023 to April 2024; bottom panel daily average temperature data obtained from
the Vaisala AWS310 weather station from June 2023 to April 2024 b).**

## 5.3 Dust intrusions

During summer and spring, the site is regularly affected by Saharan dust intrusions (Mona et al., 2006). Desert dust particles

have many effects: they can impact climate, the precipitation cycle, and human health (Sokolik et al., 2007; Mona et al., 2023).

Mineral dust particles can act as cloud condensation nuclei (CCN) and thereby determine the concentration of the initial

droplets, albedo, precipitation formation, and lifetime of clouds (Levin et al., 1996; Levin et al., 2005).

The multiwavelength polarisation Raman lidar provides highly resolved spatial and temporal atmospheric profiles that allow

for the separation of the different aerosol layers (Pappalardo et al., 2004b; Papagiannopoulos et al., 2018). Large and irregular

shaped Saharan desert dust particles produce medium lidar ratios S, relatively high PLDR values and they are spectrally neutral

to backscatter and extinction producing low Ångström exponent referred to the wavelengths 355-532 nm (Freudenthaler et al.,

2006; Fernandez et al., 2019). In fact, mineral desert dust aerosols predominantly consist of coarse mode particles of irregular

shapes (Mahowald et al., 2014).

In-situ measurements, in case of sedimentation events, provide complementary information on the advected dust. Low values

of nephelometer-derived SAE that indicate carse particles and, conversely, high aethalometer-derived AAE values that

demonstrate the wavelength dependent absorption (Cazorla et al., 2013).

When the atmosphere is dominated by particles with large dimension such as dust particle, the sedimentation is fostered and

involves higher return to the ground level so the measurements of size distribution of coarse particles by APS plays a key role

in dust studies Furthermore, the low $PM_{2.5}/PM_{10}$ ratio obtained by real time measurements using the PMx monitor could be

the confirmation that the main component of the desert dust events is the PM coarse fraction. Finally, the $PM_{10}$ mass

concentration collected over 24h on filter measured by the PMx sampler (SWAM 5a-Dual Channel Monitors, FAI Instruments)

will be higher during a dust event compared to non-dust events, with $PM_{10}$ concentration values that could exceed the European

daily limit value (2008/50/CE European directive).

Regarding the chemical characterization, the ICP-OES plays a key role in evaluating the influence of the transport of dust by

detecting the elemental composition of the mineral fraction. In particular, monitoring the concentrations of the typical crustal

elements such as As, Al, Ca Cr, Cd, Co, Cu, Fe, K, Mn, Mo, Na, Ni, Pb, Sb, Se, Sn, and Zn and Rare Earth Elements (REEs)

is relevant because are generally markedly higher during desert dust event than in comparison with their annual means (Aydin

et al., 2012; Rodriguez-Navarro et al.,2018; Mărmureanu et al., 2019).

In the following we report an example of aerosol remote sensing and in-situ observation for a Saharan dust intrusion at CIAO

to demonstrate the possibilities for synergistic combination of data from lidar and in-situ aerosol measurements. Even if only

the APS instrument was available at that time, the presence of just one in-situ instrumentation already shown the importance



of such combination of techniques. The observations are related to the second half of June. Figure 9a reports the fine mode
fraction as retrieved from CIAO photometer measurements and available at aeronet.gsfc.nasa.gov. This parameter provides
information about the fraction of fine mode particles respect to the coarse one as obtained from the AOD (Aerosol optical
Depth) measurements. This parameter is retrieved from columnar measurements and therefore refer to the total atmospheric
column. Fig 9a clearly shows that in the 20-23 June period the coarse particles are more abundant respect to previous and
following period. For the same period Hybrid Single-Particle Lagrangian Integrated Trajectory (HYSPLIT) backward
trajectories ending over Potenza indicate Sahara desert as potential source of the observed particles.
Lidar obseravtions provide a better insight of the temporal and vertical distribution of the aerosol at CIAO on those days.
Figure 9b-c report available Lidar observations for the period. It shows the color-maps of the vertical distribution and temporal
dynamics of the aerosol as time series of range-corrected lidar signals at 532 nm for the night of 22 June 2023 and the daytime
day of 23 June 2023. In particular, these plots report the component of backscattered signals at 532 cross-polarized respect to
the emitted laser light: the presence of high cross-polarized backscatter signals is a signature of presence in that portion of 4d
atmospheric region of aspherical particles, like Saharan dust ones.
The representation of the aerosol distribution during the night of 22 June (Fig. 9b) shows two main layers of dust: one at an
altitude close to 1 km above ground level (agl) and a second denser one above it at a height of approximately 3 km agl.
Particularly interesting for the potential link with in-situ measurements is a branch of the lower layer around 01:30 in the night
between 22 and 23 June, which seems to descent in altitude and could potentially sediment at the ground. It is worth to note
that the lidar blind region for the instrument available at the time of the measurements was around 400 m not allowing to
further investigate this point. Over the next day (Fig. 9c), the color-map again indicates the presence of dust from 9:00 to 12:00
at similar heights to 22 June but with lower density until it disappears after 12:00.











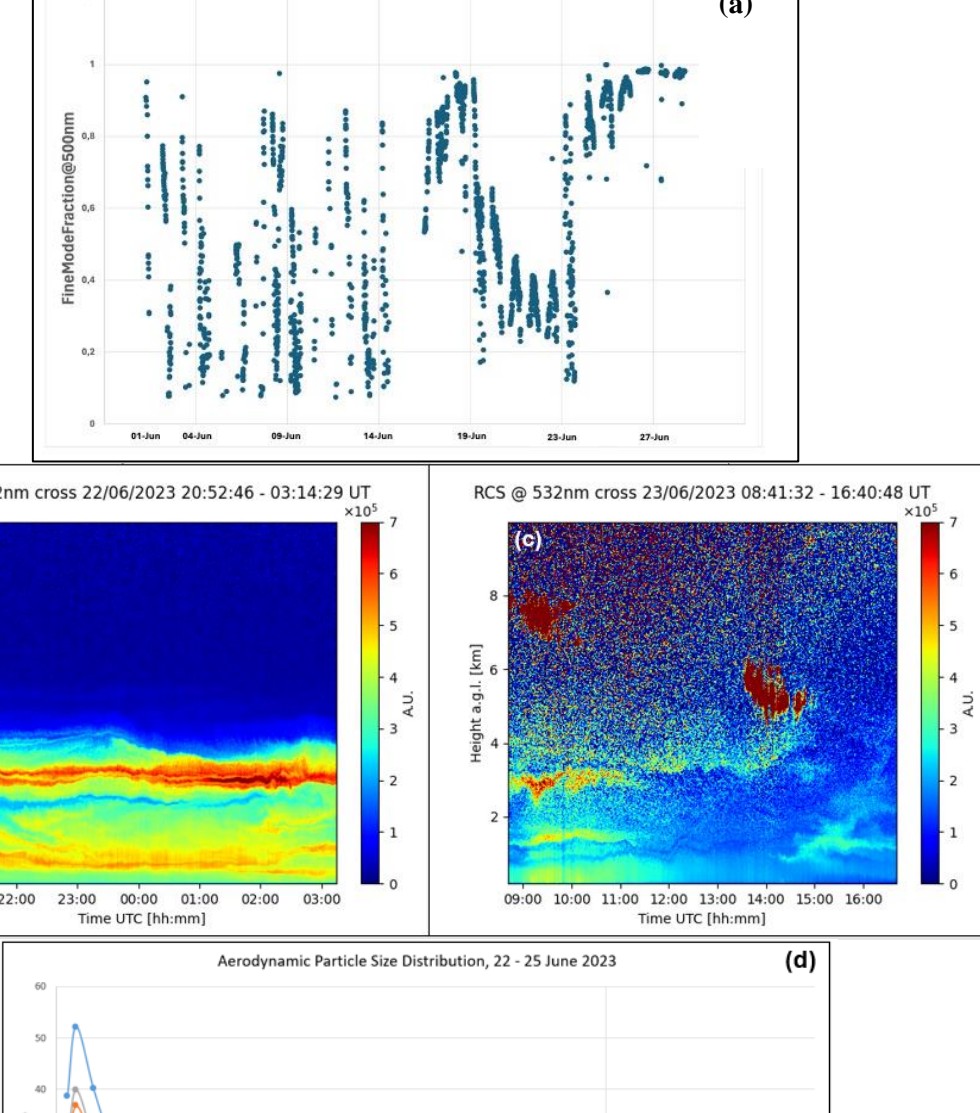

**Figure 9: Fine mode fraction as retrieved from CIAO photometer measurements related to the second half of June (a), color-coded**
**time series of range-corrected lidar signals measured at 532 nm cross-polarized channel obtained with the MUSA lidar system on**
**22 June 2023 (b) and 23 June 2023 (c), aerodynamic particle size distribution daily averages obtained with APS on 22-25 June 2023**
**(d).**





Online observations at the ground allow a better understanding of the dust presence at the surface exploring also the status
after the 22-23 June. In that period, only APS and aethalometer were operational at CIAO.
Fig. 7d shows the aerodynamic particle size distribution daily averages obtained with APS on 22-25 June 2023 and provides
complementary information to that obtained through lidar and photometer measurements. Indeed, Fig 7d distinctly illustrates
that there is negligible variance in the concentration of ultrafine particulates between dust (22-23 June) and non-dust (24-25
June) days, instead there is a noticeable rise in the concentration of fine and coarse particles with a diameter of up to 5 µm on
the dust days (22-23 June) compared to the non-dust days (24- 25 June); demonstrating how during dust events the atmosphere
is dominated by large particles (Fig 7a) distributed over different altitude ranges (Fig 7b-c) and if sedimentation is favoured,
this leads to a greater return to ground level in the coarse mode (Fig. 7d).
More information would be needed for a deeper investigation of such kind of event, and this is the reason why the CIAO
observatory has been extensively upgraded as described in this paper and we surely will observe in the next future other events
to be analysed through online and offline instruments. However, this kind of detailed investigation is out of the scope of the
current paper and would be object of further publications.

**6 Conclusions**


The recent upgrade of CIAO with the aerosol in-situ laboratory aims to provide comprehensive data on aerosol composition
and properties, which will contribute to improve climate change models and understand the effects on human health and
ecosystems. The aerosol in-situ laboratory has started in November 2023 the ACTRIS labelling process in order to prove the
operational capacities of the National Facility in ACTRIS and ensure the high quality of ACTRIS data in order to obtain the
label "ACTRIS National Facility" for the aerosol in-situ component.
The continuous in-situ measurements in tandem with the aerosol remote sensing suite will provide a valuable record of aerosol
observations for synergises. Additionally, an ICOS Atmospheric site is under implementation: this will furthermore enhance
CIAO's observing capabilities and synergies. All data collected are open and available to external users through international
databases (e.g., ACTRIS and ICOS) or through CIAO local services (e.g., meteo data). CIAO also offers remote and physical
access to the facility (https://ciao.imaa.cnr.it/access-2/), hosting researchers, students, SMEs and stakeholders, but even the
possibility to host user's instruments or sending CIAO mobile platforms to users' sites. All the above are implemented with
the main objective of fostering the advancement of the knowledge in the atmospheric field, through the wide use from the
scientific community of such extended CIAO observational datasets.
The CIAO aerosol in-situ laboratory has been built following ACTRIS suggestions and requirements, for which technical
solutions and schemas are here reported. The instrumental set up will allow to address main research topics such as the aerosol
typing and the characterization of the PBL. A first step towards integrating CIAO's different observing platforms is planned
during an extensive CIAO measurement campaign focused on the estimation of the PBL using aerosol lidar methodologies
and its validation with independent measurements and techniques that will be held in Spring 2024. Furthermore, the next-to-



come ICOS Atmospheric Class 1 site at CIAO (first step of labelled process already passed) will offer other possibilities of
synergistic studies and integration among Ris in the environmental filed. In this direction, CIAO is deeply involved in the
developments of ITINERIS (Italian Integrated Environmental Research Infrastructures System), an overarching National
project for enhancing the interlinkages of all the Italian Ris in the environmental domain. The multi-platform and multi-
disciplinary approach of the observatory coupled with the open data and open access philosophy is key for better addressing
complex atmospheric and environmental questions posed by climate change and anthropization processes.

**Author contributions**

TL, AM and ST contributed to Writing – original draft preparation. TL, AM, ST, CCol and MM contributed to Visualization.
TL, AM, ST, FC, DA contributed to Methodology. TL and MM contributed to Formal analysis. DA, AG, CD, ER and CC
contributed to Resources. CCor, SG, RMAP, GP contributed to Funding acquisition. EL contributed to Software. TL, AM, ST,
AA, BDR, MR, LM contributed to Writing - review & editing. FC, EL, FM, DA, AG, CCor, BDR, CD, SG, MM, NP, GP,
RMPA, ER, DS and CCol contributed to Review & editing. GP and LM contributed to Conceptualization. LM contributed to
Project administration and Supervision.

**Competing interests**

The authors declare that they have no conflict of interest.

**Acknowledgements**

The authors acknowledge the MIUR (Italian Ministry of University) PON Ricerca e Innovazione 2014-2020 – PER-ACTRIS-
IT – "Potenziamento della componente italiana dell'Infrastruttura di Ricerca Aerosol, Clouds and Trace Gases Research;
CIR01_00015 - PER-ACTRIS-IT "Potenziamento della componente italiana della Infrastruttura di Ricerca Aerosol, Clouds
and Trace Gases Research Infrastructure - Rafforzamento del capitale umano" - Avviso MUR D.D. n. 2595 del 24.12.2019
Piano Stralcio "Ricerca e Innovazione 2015-2017" and CIR01_00019 - PRO-ICOS_MED "Potenziamento della Rete di
Osservazione ICOS-Italia nel Mediterraneo - Rafforzamento del capitale umano" - Avviso MUR D.D. n. 2595 del 24.12.2019
Piano Stralcio "Ricerca e Innovazione 2015-2017".The authors also acknowledge the IR0000032 – ITINERIS, Italian
Integrated Environmental Research Infrastructures System (D.D. n. 130/2022 - CUP B53C22002150006) Funded by EU -
Next Generation EU PNRR- Mission 4 "Education and Research" - Component 2: "From research to business" - Investment
3.1: "Fund for the realisation of an integrated system of research and innovation infrastructures" and ATMO-
ACCESS (Access to Atmospheric Research Facilities) Funded in the frame of the programme H2020-EU.1.4.1.2 – Grant



Agreement n. 101008004 – (1 April 2021 – 31 March 2025)..The authors also acknowledge the Joint Research Unit ACTRIS-
Italy funded by the Italian Ministry of University and Research.

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
