# Peer review of "CIAO observatory main upgrade: building-up an ACTRIS compliant"

_Atmospheric Measurement Techniques, 2024_

## Author Comment (AC1)

[Figure]

**Figure 3: Wind rose diagram at CIAO in 2004-2017 obtained from continuous measurements of the automatic weather station VAISALA MILOS520 with temporal resolution of 1 minute.**

---

## Author Comment (AC2)

**Response to comments from Anonymous Referee #1**

**We thank the reviewer for taking the time to read the manuscript and provide detailed and valuable feedback.**

The above manuscript describes the set-up and characteristics of a combined aerosol in situ and remote sensing measurement side, operated by CNR in ACTRIS. At the beginning, I had my doubts if AMT is the right place for this. But there are, at least for me, two strong arguments, why the manuscript should become an AMT publication. Firstly, for each infrastructure, a reference paper is needed, where, hopefully, all the scientific papers to come can refer to. Secondly, as the authors claim, such a paper can act as "a practical guide for implementation", in particular for researchers in America or Asia, where ACTRIS is probably not so well known. However, to make these two arguments valid, some more detailed information must be provided that the paper can act as reference. And if it should be a practical guide for implementation, for me it is a must to give at least an overview about associated resources (time, man-power, maintenance costs, etc.), both concerning the implementation as well as for the operation later on. This would a with ACTRIS not familiar person allow to do a cost benefit analysis. It would be interesting to go further into that direction and give an estimation on how much of these stations would be needed across Europe or globally to cover the scientific needs. But this is only a nice-to-have remark, no request.

We thank the reviewer for this comment that helped us to improve the paper in order to better underline the importance of the paper for extra European /outside ACTRIS community. Indeed, the ACTRIS aerosol in-situ standards are in some way following the GAW ones, therefore the interest in technical solutions for an ACTRIS compliant in-situ instrumentations stays not only with stations potentially involved in ACTRIS. More relevant for European perspective, new air quality directive is under definition and this legislation is taking into account standards developed into ACTRIS for example for black carbon (BC) and ultrafine (UF) particles. The approach of such new EU air quality directive is to have BC and UF measurements in some more advanced stations of the National Air Quality management system. Solutions adopted for collecting such measurements with ACTRIS standard could be of interest for AQMN for guarantying the quality of collected data.

A short sentence about the potential wide interest of our paper outside ACTRIS at European and International level will be added in the revised paper.

About resources associated to the CIAO aerosol in-situ component, we estimated a total initial investment of about 1 M€, and 2 years were needed for building it up from scratch. For the operation, we estimate about 30k€ as maintenance and 70€ as consumables as annual operational costs. In addition, at least 2 researcher fulltime is needed for running this in-situ instrumentation with the support of a technicians (half time).

Such indications are shortly provided in the revised version of the paper.

How many of such stations are needed to cover the scientific needs: very important question to which is difficult if not impossible to reply to. The geographical density (and the location) of needed number of stations depends on the specific scientific question. Aerosols are highly variable in space and time and therefore measurements collected in different places can strongly differ. For air quality issues, aerosol in-situ measurements should be collected where more inhabitants are for investigating the impact on health, but for decoupling the background levels from extremely local source (e.g. the single car),

background stations are needed. If instead the scientific aim is understanding how much the EU policies allow the reduction of PM, few stations maybe just one in remote site could be sufficient.

GAW indeed makes a distinction between: **Local stations** for research and supporting services related to urban environments, and in other locations impacted by nearby emissions (e.g. from biomass burning); **regional stations** for which the station location is chosen such that, for the variables measured, it is regionally representative and is normally free of the influence of significant local pollution sources or at least frequently experiences advection of pollution-free air from specific wind directions; finally **global stations** primarily observe GAW variables under background conditions, i.e. without permanent significant influence from local pollution sources (https://wmoomm-my.sharepoint.com/:w:/g/personal/jbourdeu_wmo_int/EXLkdQpkP-RBi1ooPA3zXpkBHxUFFhX25aJTnj4L4b8k6Q?e=xdra6a). Even if with this classification, the GAW implementation plan is not currently providing a clear directive related to the number of stations and currently GAW accounts for 8 Global stations over Europe, while over Northern America only 2 (https://community.wmo.int/en/activity-areas/gaw/research-infrastructure/gaw-stations/gaw-global-stations) .

In ACTRIS an intermediate solution is adopted, trying to have enough stations in each participating country to cover different regional characteristics. In Italy for example we have aerosol in situ site in Po Valley (north and polluted area), Potenza as background site in the Apennine (south), Lecce Southern Italy a city on the seaside, and Rome as big metropolitan area in the central Italy.

**Specific remarks:**

1. p. 1, l. 13: CNR, IMAA and all the other abbreviations in the manuscript. Please write the full name, when you us the acronym the first time. And as the manuscript has so many acronyms, please add a glossary.

   We will ensure that all abbreviations, including CNR-IMAA, are spelled out in full the first time they are mentioned in the manuscript. Additionally, we will include a glossary to help clarify all acronyms used throughout the text.

2. p. 1. l. 18: I´m not so sure if the provided examples are examples for a "synergistic approach", both approaches, remote sensing and in situ, a complementary and hence help each other to get the full picture. My understanding of synergy would be that a combination of these two methods provide a totally new aspect, which cannot be obtained by one method alone. But please convince me that I´m wrong.

   When we talk about synergy, we refer to the ability to reveal new phenomena or insights that emerge only using together the 2 types of observations. Other researchers (e.g. Davulien et al., 2023 doi.org/10.1016/j.scitotenv.2023.167585) intended synergistic approach in a similar way. One example of this type of "synergistic approach" is related to the wintertime aerosol particle conditions. During winter, low clouds and fog often occur at CIAO and therefore lidar measurements are inhibited or no aerosol optical properties can be retrieved from lidar measurements.
   In small number of cases, aerosol particle properties profiles are obtained by lidar measurements in winter.

The climatological profile of aerosol backscatter at 532 nm  for winter season  2000-2019 at CIAO (https://doi.org/10.57837/cnr-imaa/ares/actris-earlinet/level3/climatological/2000_2019/pot) shows very clean air respect to other seasons in the whole investigated atmospheric column.

Only the last point close to the surface is slightly higher, but the information content is too low for further investigation. These cases are typically considered as clean day from the aerosol remote sensing perspective.

[Figure]

But it should be considered that the lidar is blind in the lowest portion of the atmosphere and it is expected that due to the low BLH, most pollutant stays within the low BL area.

Aerosol in situ measurements instead do no see above the boundary layer height, but well capture what's happening close the surface. In wintertime the BC is higher at our site probably because of the increase in using heating system (typically fireplaces). Only such in-situ observations allow to understand that winter cases are not to be considered as background conditions below the boundary layer height.

On the other side, remote sensing excels at capturing large-scale, vertical, and temporal variations in aerosol distributions (e.g., the spread of wildfire smoke or desert dust layers), while in-situ instruments are able to offer a detailed chemical composition, size distribution, and ground-level concentrations. This synergy enables scientists to not only track dust but also assess its immediate and long-term impacts on ecosystems and human health, which would be unattainable with either method alone.

The unique value here comes from being able to validate and interpret remote sensing data using in-situ measurements, and vice versa, creating new insights that wouldn't be possible with only one method. For example:

While remote sensing and in-situ approaches are complementary in nature, the true synergy emerges when these methods work together to offer new insights and reduce uncertainties that cannot be achieved by one alone. This synergy is essential in environmental monitoring, as it helps create more accurate, multi-dimensional views of atmospheric processes, ultimately leading to better scientific understanding and policy decisions. Another aspect of synergy lies in reducing uncertainties in models. Aerosol-climate models, for instance, rely on accurate

ground truth data. Remote sensing gives a broader atmospheric overview, but it can suffer from calibration issues or may lack the fine details on chemical composition. In-situ data can calibrate and validate these remote measurements, ensuring that models predict both the spread and chemical impacts of aerosols more accurately. This is a completely new layer of certainty that one method alone cannot provide.

Some revisions will be done on the paper to tackle the reviewer's comment.
Firstly, based on reviewer's comments we decided to adopt combined instead of synergistic word in the text for what has been done up to now, but leaving the concept of synergistic approach for the further developments that we imagine and plan for the near future, but that are not yet achieved. The current paper is focusing on the implementation of the instruments in view of such synergistic approaches, which will be object of further paper(s).
Secondly the cases section will be reviewed (see reply to comment 23).

3.  p. 1 l. 28: Pöschl, 2005 reference is fine, but two decades old. Maybe add also a newer reference?

    We will add the following more recent references to the manuscript to complement the Pöschl (2005) reference:

    **IPCC (2021).** *Aerosols and their impact on climate and human health.* In *Climate Change 2021: The Physical Science Basis.* Contribution of Working Group I to the Sixth Assessment Report of the Intergovernmental Panel on Climate Change (IPCC). Cambridge University Press.

    Ren-Jian, Z., Kin-Fai, H., & Zhen-Xing, S. (2012). The Role of Aerosol in Climate Change, the Environment, and Human Health. *Atmospheric and Oceanic Science Letters*, *5*(2), 156–161. https://doi.org/10.1080/16742834.2012.11446983

4.  p. 2 l. 34: I miss the refrence to the ACTRIS BAMS overvies paper here

    The reference to the ACTRIS BAMS overview paper has been added to the manuscript:

    "Laj et al Aerosol, Clouds and Trace Gases Infrastructure (ACTRIS); The European Research Infrastructure Supporting Atmospheric Science, BAMS, E1098-E1136, https://doi.org/10.1175/BAMS-D-23-0064.1, 2024."

5.  p. 2 l. 56: the "labelling process" is well known for ACTRIS people. But already European users are unfamiliar with this, not to speak about people from other continents. As the labelling process is an important part of the data quality assurance in ACTRIS, it should be **shortly** described what it is for and how it works, and for the details reference should be given.

    I agree with your point, and we will add the following brief description along with a detailed reference:

    "The "labelling process" in ACTRIS is a key element of its data quality assurance system. This process ensures that instruments, data, and methodologies used across ACTRIS observational platforms meet specific quality criteria. The labelling process involves a series of evaluations and certifications to verify compliance with ACTRIS protocols."

Reference: Deliverable 5.1: ACTRIS NF Labelling Plan,
https://www.actris.eu/sites/default/files/Documents/ACTRIS%20IMP/Deliverables/ACTRIS%20IMP_WP5_D5.1_ACTRIS%20NF%20Labelling%20Plan.pdf

6. p. 3 l. 67: "reference observatory for atmospheric research"? Who claims this? A reference for atmospheric dynamics and ozone hole chemistry? Probably not. I can imagine that it is "a reference station for short-live atmospheric constituents in Italy and the Mediterranean".

   You are correct, and I appreciate your insight. The term "reference observatory for atmospheric research" may be too broad and misleading in this context. It would be more accurate to describe it as "could be a reference station for short-lived atmospheric constituents in Italy and the Mediterranean."

   This change will clarify the specific focus of the station and better reflect its role in monitoring atmospheric dynamics and constituents relevant to our studies. Thank you for bringing this to my attention; the modification will help ensure that the description is precise and aligns with the station's actual contributions to atmospheric research.

7. p. 3 photo: a lot of infrastructure, all of this is in situ aerosol? Better to zoom in and show the in situ aerosol containers and the inlets.

   In the photo, we are showing the various infrastructures of the CIAO site, which provide a comprehensive view of the facilities. However, we agree that it would be beneficial to include a zoomed-in photo specifically highlighting the in-situ aerosol containers and the inlets. This will provide a clearer understanding of the aerosol sampling setup. We will make sure to add that zoomed-in image to enhance clarity and focus on the relevant components.

8. p. 5 l. 122: you state to list the "research lines" in the following rows here. But "development of", "implementation of", "harmonization of" etc. are no research lines, they are intended steps to allow your research later on. Hence the simplest way to make this consistent would be to replace "research lines" here with a more adequate wording.

   You make a valid point regarding the terminology used. Terms like "development of," "implementation of," and "harmonization of" are indeed steps or processes rather than distinct research lines. To improve clarity and consistency, we will replace "research lines" with "research themes" or "research objectives."

9. p. 6, table 1: the list of instruments is compelling, but, as an in situ person, I would rather like to know which parameters are provided by the remote sensing devices. This would be a good suggestion anyhow, make a table providing all the remote sensing and in situ parameters at the same spot, thus the potential synergy gets more visible.

   In the revised manuscript, we will update the table to include both the remote sensing and in situ instruments, along with the respective parameters they measure. This will help to better highlight the potential synergy between the different measurement techniques.

10. p. 6 l. 144: "Observational Platform" and "Exploratory Platform" are known to some of us, but not to the waste majority of the readers. Please explain shortly (in parentheses) what these terms stand for.

    We will add the following explanations in parentheses in the manuscript to clarify the terms:

Observational Platforms (fixed ground-based stations delivering long-term high-quality data and continuous atmospheric monitoring on a regular schedule and common operation standards by applying state-of-the-art remote-sensing and in situ measurement techniques.)

Exploratory Platform (atmospheric simulation chambers, laboratory platforms and mobile platforms that perform dedicated experiments and contribute data on atmospheric constituents, processes, events or regions.)

11. p. 6 l. 152: The description of the Central Facility part of CIAO is surely correct, but not needed for the purpose of the paper and rather confusing for the reader. Please omit this here, it is better described elsewhere.

We will eliminate or condense the description of the Central Facility part of CIAO in this section, as it is not essential for the purpose of the paper and may cause confusion.

12. p. 9. L 211.: which heads do the PMx instruments have? Please give this information.

This information is already provided in section 4.4, which details the instrument configuration. However, to clarify it here as well:

"Additionally, two PMx samplers (SWAM 5a-Dual Channel Monitors, FAI Instruments) are installed with respective inlets: one equipped with two PM2.5 inlets, and the other with one PM10 and one PM1 inlet. Furthermore, a PMx monitor (EDM 180, Grimm) is placed as a standalone instrument with individual PM10 inlet line."

13. p. 9 l. 218: the isokinetic flow splitter: which one did you use? Or can you provide as drawing of it?

The isokinetic flow splitters were custom-built to meet ACTRIS requirements by "4S SOLUZIONI E SVILUPPO PER LA STRUMENTAZIONE SCIENTIFICA," a company specialized in metrology, measurement physics, and the development of scientific instruments, with a particular focus on atmospheric observation tools. In the manuscript, we will include photos to provide a detailed view of the splitters.

14. p. 9 l. 219: I have some experience with sampling lines, but I only can guess the argument of the sharp tube ends, please be more specific.

The ends of the tube in the isokinetic flow splitter must be sharp to ensure a homogeneous distribution of the air sample because sharp edges help minimize flow disturbances and turbulence at the entrance of the sampling tubes. A sharp edge promotes a smooth transition of the air flow, reducing the risk of vortices and irregular flow patterns that can lead to uneven sampling.

When the air enters the tubes with a well-defined, sharp edge, it helps maintain the laminar flow conditions that are crucial for accurately capturing the aerosol particles. This ensures that the sampled air maintains a consistent velocity and composition, allowing for representative measurements and reducing the potential for aerosol losses due to inertial effects or diffusion. In essence, a sharp tube end facilitates better mixing and uniformity in the airflow, which is vital for the accuracy and reliability of aerosol measurements.

The following revised sentence will be added to the revised version of the paper:

"Moreover, the tube ends in the isokinetic flow splitter must be sharp to minimize turbulence and promote smooth airflow, ensuring uniform sampling. This design helps maintain laminar flow, reduces aerosol losses, and enhances the accuracy and reliability of measurements."

15. p. 9 table 2: first of all, the two first Reynolds numbers are equal, even if the flow rate is different. The upper one is wrong in my opinion. Same for the speed there.

To get a feeling about the sampling lines could you please add line length and number of bends?

There was indeed a typo error in the table regarding the flow rate of the aethalometer, which is not 3 l/min but 5 l/min. As a result, both the aethalometer and the nephelometer, having the same internal diameter of the isokinetic splitter and flow rate, also have the same Reynolds number and flow speed. This will be corrected in the manuscript.

We will include the information regarding the line length and number of bends in the revised version of the manuscript.

16. p. 10 l. 228: for me one of the most critical points. Knowing from the literature and also from own experience, conductive "plastic" tubes can be critical, both concerning particle losses as well as chemical composition. The chosen tube MIGHT be OK, but please either provide a reference for that or provide own measurement data e. g. on the size-resolved particle transmission. Otherwise all your data are always "conditionally" correct only.

We sincerely thank the reviewer for highlighting this critical point. In the manuscript, there was an incorrect definition regarding the material of the tubes used. The tubes employed are actually TSI sampling black tubes. These TSI sampling tubes are made from conductive silicone, infused with carbon black to enhance conductivity. This design is essential for minimizing electrostatic losses, which can occur in non-conductive tubes, such as those made from standard silicone or Teflon, where particles may adhere to the tube walls due to static charges. The conductive nature of TSI tubes prevents the buildup of electrostatic fields, thereby improving particle penetration and reducing sampling biases caused by particle loss. These tubes are recommended and considered the best for particle sampling. Further details will be provided in the revised version of the manuscript.

17. p. 10 l. 232: the Nafion dryer, which one? Please provide reference or explain, what this is

A **Nafion dryer** is a specialized device used in aerosol sampling to remove water vapor from a gas stream while preserving the integrity of the sample's chemical composition. Nafion membrane is a sulfonated tetrafluoroethylene-based polymer that selectively transfers water vapor from a gas sample to a surrounding purge gas, while retaining the sample's other gases and particles. This property makes it ideal for removing moisture from aerosol-laden air without affecting the aerosol particles themselves.

We will include the following reference in the manuscript regarding the Nafion dryers we use: MD-700 Series Dryer for Aerosol Analysis.

18. p. 10 l. 240: why is the 2:1 flow ratio desired, please explain

The 2:1 purge-to-sample flow ratio in a Nafion dryer operate in reflux mode is crucial for achieving efficient aerosol particulate sampling. This ratio ensures there is enough dry purge gas to continuously absorb moisture from the sample, which prevents the purge from becoming

saturated. By maintaining this higher purge flow, the system can keep moisture levels low, ensuring that the sample's integrity is preserved. This is particularly important for aerosol particulate sampling, where even small amounts of moisture can alter the particle characteristics and lead to inaccurate measurements. The 2:1 ratio helps maintain consistent drying efficiency over time, which is essential for reliable aerosol analysis.

Here is the revised version that we will include in the manuscript with the shortened explanation:

"The vacuum on the purge air should be at least 15 inches Hg, with a higher vacuum preferable. This vacuum level is required to provide the desired 2:1 purge-to-sample flow ratio based on the actual volumetric flow. The 2:1 ratio ensures enough dry purge gas to continuously absorb moisture, preventing saturation and preserving sample integrity. This is crucial in aerosol particulate sampling, where even small amounts of moisture can affect particle characteristics and compromise measurement accuracy."

19. p. 13 l. 317: "unattended measurements ... on the timescale of years" is an overstatement, you have to check the instruments regularly, even if they might be OK for one or the other year (which I personally doubt). Please soften this statement.

   We agree that the statement could be softened. While the robustness of the instrument was the point being emphasized, it is indeed essential that the instruments are regularly checked and calibrated to ensure the accuracy and reliability of long-term measurements. We will adjust the wording to reflect this and avoid the implication that the instruments can operate indefinitely without maintenance.

20. p. 13 l. 338: the statement that Potenza is a rural site is a trivial statement, please remove or phrase differently, what you want to highlight

   We will remove the statement referring to Potenza as a rural site, as it might be too simplistic. Here's a rephrased version of the initial sentence:

   "As a general comment, we could say that the Potenza site exhibits low PM concentrations and a very high contribution of the organic substances, as observed in rural areas."

   p. 15 l. 366: in the manuscript, I miss some more evaluation data, checking the consistence of the measurements e.g. here, how good the mass measurements of the different instruments agree with the size distribution derived mass etc.

   The reviewer is perfectly right. Several tests can be made for cross checking the instruments (between the aerosol in-situ instruments as well as versus the aerosol remote sensing ones). Anyhow the aim of the current paper is the description of the different steps and solution adopted for implementing such a large aerosol in-situ laboratory coupled to an existing remote sensing observatory. Part of the checks suggested by the reviewer are part of the quality assurance procedures working in ACTRIS and we are proceeding with those. Additional cross checks will be done taking the most from the plethora of CIAO instruments and expertise. However long record of data would be desirable for an assessment of the consistency and accuracy of the measurements. The results will be included and fully explained in subsequent manuscripts.
   A short sentence about this will be added in Final section of the revised paper.

21. p. 15 l. 370ff: The elemental analysis, what is given in ACTRIS there or is this just an add-on to the in situ aerosol particle properties?

The elemental analysis of aerosol particles is a recommended variable by ACTRIS, though it is not mandatory for a site to become an ACTRIS National Facility (NF) observatory. Despite not being compulsory, elemental analysis of in situ aerosol particles plays a crucial role in understanding the composition and sources of atmospheric aerosols, as well as their impact on air quality, health, and climate.

The elemental analysis provides important data on the concentration of potentially toxic elements and trace metals in particulate matter, helping to distinguish between natural and anthropogenic sources. This type of analysis is essential for more detailed studies on the environmental and health impacts of aerosols.

For this reason, we have equipped our site with ICP-OES (Inductively Coupled Plasma Optical Emission Spectroscopy) and an OC/EC analyzer (Organic Carbon/Elemental Carbon) to perform comprehensive elemental and carbon content analysis, which significantly enhances our capability to assess aerosol properties in detail.

In the revied paper we add a short sentence at beginning of section 4.1 about mandatory and recommended instruments for ACTRIS aerosol in situ facility.

22. p. 17 l. 433: the three cases: I believe I understood what the authors wanted to show here, but I have the feeling that most of the statements can be already given with only one of the two methods, in situ and remote. I do not see the real synergy. This would be the case, at least for me, if you would use in situ and remote sensing data to generate another data product. Please elaborate a little bit more on that section, otherwise you weaken your own argument that collocated measurements are valuable. At the same time please shorten the section, there is too much "text book" knowledge in.

This comment is linked to the point 2 above, where we discussed the general framework. Here the comment is more specific to the three show cases. As reported in the paper, "we present three emblematic cases recurring at CIAO where the combined deployment of the in- situ and remote sensing observations is expected to be of added value". Here we describe the 3 topics on which we expect the most from the, combined first and synergistic then, use of remote sensing and in-situ data. The description of these topics allows also to report first 10 months record of black carbon measurements collected at CIAO and a show case of desert dust arrival over CIAO captured by the lidar and photometer measurements and the identification of its intrusion down to the ground by aerosol in-situ measurements.

To address the reviewer comment, for the three cases we will shorten the section by reducing the textbook-like knowledge and focus more on the added value of combining these measurement techniques. In particular: 1) local wild fire subsection will be mentioned as potential case of investigation in the text; 2) winter pollution: paucity of aerosol remote sensing, low BLH and blind lidar region for this period and needs of independent and in-situ measurements for such period relevant for air quality (health) issue will be underlined; 3) desert dust: it will be better underlined how such case demonstrates firstly the agreement between the 2 observations in capturing a desert dust event, underlining how each one of the technique overcomes the limit of the other one and finally we will describe potential future synergistic products.

Moreover, I believe modelling could strongly benefit from collocated in situ and remote sensing measurements, but this is not addressed in the manuscript.

This suggestion will be addressed in the revised version of the manuscript by highlighting the added value of the availability of in situ collocated measurements and remote sensing of aerosols in increasing the accuracy of model predictions, allowing the reduction of uncertainty of aerosol measurements in the atmosphere (e.g., Vratolis et al., 2020), as well as in the evaluation of aerosol models. Furthermore, in recent years, the use of collocated aerosol measurements has found application in training machine learning-based models (see, e.g., Redemann and Gao, 2024).

Redemann, J. and Gao, L.: A machine learning paradigm for observations needed to reduce uncertainties in aerosol climate forcing, Nature Communications, 15, 8343, 2024.

Vratolis, S., Fetfatzis, P., Argyrouli, A., Soupiona, O., Mylonaki, M., Maroufidis, J., Kalogridis, A.-C., Manousakas, M., Bezantakos, S., Binietoglou, I., Labzovskii, L.D., Solomos, S., Papayannis, A., Močnik, G., O'Connor, E., Müller, D., Tzanis, C.G., Eleftheriadis, K.: Comparison and complementary use of in situ and remote sensing aerosol measurements in the Athens metropolitan area, Atmos. Environment, 228, 117439, 2020.

A small paragraph about this aspect as well as about the relevance of aerosol insitu measurements for qualifying the microphysical properties retrievals based on remote sensing measurements will be explicitly reported in the revised version.

23. p. 24. l. 620: the conclusion is not a conclusion, it is mainly written as outlook. Outlook is fine in at the end of the conclusion, but should not cover the major part of the text. Hence please rephrase the conclusions. Therefore imagine, e.g. the three most important statements the reader should have learned reading your paper.

The conclusions in the revised manuscript will be as follows, highlighting the most important statements the reader should have learned from reading the paper:

"The recent upgrade of aerosol in-situ laboratory to the well-established remote sensing activities at the CIAO observatory significantly enhances its observational capacity. The integration of in-situ and remote sensing measurements offers a more complete understanding of aerosol behaviour, enabling detailed studies from ground level up to the stratosphere. This combination adds value by providing both vertical profiles by remote sensing measurements and precise ground-level chemical and physical properties through in-situ measurements, which is crucial for improving climate models and understanding aerosol impacts on human health.

Establishing the aerosol in-situ facility has been a complex and labour-intensive endeavour. The process, which began in 2018, required careful planning, technical expertise, and collaboration with field specialists. The setup involved designing and implementing ACTRIS-compliant inlets, sampling lines, and advanced instruments to ensure accurate and reliable measurements. This development highlights the significant effort required to meet international standards and provide high-quality data for the scientific community.

Given CIAO's strategic location in the Mediterranean, the case studies planned for future research are especially relevant. The site is frequently affected by Saharan dust intrusions, which impact air quality and ecosystems, and the observatory is strategically positioned to study these phenomena. Moreover, the Mediterranean is also prone to wildfires, which are projected to increase in intensity and frequency due to climate change. The CIAO observatory

can monitor both the short-range transport of smoke from local fires and long-range plumes from major events, providing insights into their effects on air quality and human health. Lastly, local winter pollution, which results from residential heating, can also be analysed in detail, particularly during temperature inversions that trap pollutants near the ground. The combination of in-situ and remote sensing measurements will help investigate these key environmental issues.

Furthermore, the next-to-come ICOS Atmospheric Class 1 site at CIAO (first step of labelled process already passed) will offer other possibilities of synergistic studies and integration among Ris in the environmental filed. In this direction, CIAO is deeply involved in the developments of ITINERIS (Italian Integrated Environmental Research Infrastructures System), an overarching National project for enhancing the interlinkages of all the Italian Ris in the environmental domain. The multi-platform and multi-disciplinary approach of the observatory coupled with the open data and open access philosophy is key for better addressing complex atmospheric and environmental questions posed by climate change and anthropization processes."

**Technical corrections:**

1.  p. 1, title: isn´t it "building-up" with a hyphen?

    it will be done

2.  p. 1 l. 21: not sure if "container organization" does fit in here, "container layout" sounds better. Also "optimization" should be deleted in l. 22 in order to make the list more homogeneous.

    it will be done

3.  p. 1 l. 26: might sound nitpicking but I believe it is important to state "aerosol particle" or "particle" everywhere, where the particles are meant, and not "aerosol", which are the particles and the surrounding gas. Please check this in the whole manuscript.

    We will modify the term **aerosol** to **aerosol particle** or simply **particle** throughout the manuscript wherever particles are specifically meant, ensuring clarity and precision in the terminology used.

4.  p. 2 l. 33: Please add "The" before "Aerosol Clouds and Tr...."

    it will be done

5.  p. 2 l. 45: here CNR-IMAA has a hyphen, on page one not. Please be consistent.

    We will ensure consistency throughout the manuscript by using **CNR-IMAA** with a hyphen uniformly. This adjustment will be made to maintain coherence across the entire document.

6.  p. 4 wind rose: you might have used all the wind speed classes shown in the legend, but in practical, only winds up to 10 m/ show up. Hence make the higher wind speed classes just one additional " and larger" bin. This also prevents that the same color shows up more than once in the legend.

    Thank you for your valuable suggestion. We fully agree with your observation, and the wind rose has been adjusted accordingly. The higher wind speed classes have been consolidated into one additional "and larger" bin to reflect the practical wind speeds observed, preventing the repetition of colors in the legend. The updated wind rose diagram is available in the supplemental PDF files under "New Wind Rose Diagram."

7. p. 5 l. 95: either "Small and Medium-sized Enterprises" or "small and medium-sized enterprises" but not a mixture of both small and capital letters.

We will ensure that the term **small and medium-sized enterprises** will be written entirely in lowercase throughout the manuscript for consistency.

8. p. 5 l. 98: delete the "and", because there still follows the "or" in the list

it will be done

9. p. 5 l. 98: please make it either "to contribute instruments," or "to contribute to the instrumentation,"

it will be done

10. p. 5 l. 100: please exchange "revolve around" with "evolved within"

it will be done

11. p. 5 l. 104: please exchange "measurements" with "data", because you provide the quality-assured data for the satellite validation.

it will be done

12. p. 5 l. 114: please exchange "smokes" with "smoke plumes"

it will be done

13. p. 6 l. 128: please add an "and " before " i)"

it will be done

14. p. 6, table 1.: "lidar and optical laboratories" are infrastructure, no "instrument" as stated in the table caption.

15. p. 6 l. 149: I learned that there should always be a space between the number and the unit, i. e. should be "20 km" here. (Only exception "10°C"). Please check the whole manuscript.

We will review the entire manuscript and ensure that there is a space between the number and the unit (e.g., "20 km"), except for specific cases like "10°C." All instances will be corrected accordingly.

16. p. 7 l. 152: should be "Central Facilities" starting with capital letters. Please check the whole manuscript.

We will ensure that **"Central Facilities"** is consistently written with capital letters throughout the manuscript. All instances will be corrected accordingly.

17. p. 8 l. 199: please make it either "particulate matter collected on filters." or "aerosol particles collected on filters."

It will be done

18. p. 8 Fig. 4 caption: please add "aerosol" before "facility"

It will be done

19. p. 9 l. 206/209: please exchange "under" with "downstream", because that is meant

It will be done

20. p. 9 l. 208: please move the comma after "3938" and remove the hyphen to be consistent

*It will be done*

21. p. 9 Fig. 4: please add the information of the inlet heads also to the PMx instruments

*It will be done*

22. p. 10 l. 244: please exchange "Instead" with "In contrast"

*It will be done*

23. p. 10 l. 244: please move the "since …" half sentence to the end of the sentence, the subject and the verb should not be separated.

*It will be done*

24. p. 11 l. 250: please exchange "input" with "inflow"

*It will be done*

25. p. 11 l. 259: please add "matter" after "particulate".

*It will be done*

26. p. 12 l. 281: "human range of visibility" sounds strange for me, maybe "the visible part of the electromagnetic spectrum" is better

*It will be done*

27. p. 13 l. 322: what is " < ng m$^{-3}$", please complete this equation or write it in words

*We will either write "< ng m$^{-3}$" in words (e.g., "less than nanograms per cubic meter"). This adjustment will be made in the manuscript.*

28. p. 14 l. 340: please exchange "increasing" with "peaks"

*It will be done*

29. p. 14 Fig. 7: please add "a)" and "b)" to the two rows of plots and change the figure caption accordingly

*It will be done*

30. p. 16 l. 415: "aerosol load" is very unspecific, please use a more appropriated term for what is meant here

*We will replace the term **"aerosol load"** with a more specific term like **"aerosol concentration"***

31. p. 17 l. 446: please exchange "in short" with "over short"

*It will be done*

32. p. 20 l. 530/532: please remove the "%" after "BB", same in fig. 8 b

*It will be done*

33. p. 21 l. 553: should be "coarse" not "carse"

*It will be done*

34. p. 25 l. 649: why some words in the "authors contribution" section are written in capital letters and others not, is not clear to me.

The words in the "Authors Contribution" section were capitalized to highlight the different contributions of the authors in line with the guidelines provided by the AMT journal. However, we can modify this and switch to lowercase for consistency if preferred. This will ensure uniformity in the manuscript's style.

35. p. 25 l. 662, acknowledgement: Most of the guidelines for setting-up an in situ aerosol side in ACTRIS are given by the in situ aerosol Topical Centre. Hence, here and also in the text this TC should be acknowledged, as many years of hard work are the basis for this.

We fully agree with your suggestion and will include the acknowledgement. We will make sure to acknowledge the in-situ aerosol Topical Centre (TC) in the acknowledgements section, recognizing the many years of hard work that have provided the essential guidelines for setting up the in situ aerosol site within ACTRIS.

---

## Author Comment (AC3)

**Response to comments from Anonymous Referee #2**

We thank the reviewer for taking the time to read the manuscript and provide detailed and valuable feedback.

The paper describes the CIAO laboratory that have been recently added an aerosol in-situ measurement component that is aimed to add to the ACTRIS RI in future.

Overall, I am a bit worried if the paper is publishable in it's current shape. On the one hand there is the title stating the in-situ upgrade. That is described but ther eis as well a large portion of general description of the station that have in part already described by some papers of some of the authors.

The processes that are linked to the compliance with ACTRIS and the procedures to achive that, like "labelling" are a very specific in frame of European research infrastructures and play a minor role outside these communities. Therefore, such terms are kind of a jargon and needed some explanation, at least.

We thank the reviewer for his/her comment that pushes us to improve our paper. We will shorten the description of the remote sensing components for highlighting the novel aspect developed at CIAO, We will also avoid the ACTRIS jargon throughout the paper for facilitating the reading for not ACTRIS readers. These 2 aspects are really valuable for improving the quality of our paper. Thanks for pointing this out.

The paper tries to fulfil AMT's requirements but that leads to a situation where the scientific strenght remains low and the bare description of a measurement station or system alone without innovation may not qualify for the journal. However, I can imagine that AMT may set rules to make descriptions of such large scale efforts and stations or systems and by that reduce the antagonistic problem these type of descriptions are causing.

The aim of our paper is **to benefit the aerosol community providing a comprehensive and detailed description of technical solutions for the implementation of such component** (aerosol in situ). The development of such laboratory for aerosol in situ measurements required designing implementing and optimizing technical solutions. This paper is the way in which we want to share our experience allowing others to do not repeat all the process but adopting and in case of needs tailoring our solutions to their needs. Examples of measurements and their combined use with aerosol remote sensing observations are reported as show cases of potentialities of CIAO extended observatory.

Such a topic perfectly fits in our opinion with AMT scope: *The main subject areas comprise the development, intercomparison, and validation of measurement instruments and techniques of data processing and information retrieval for gases, aerosols, and clouds. Papers submitted to AMT must contain atmospheric measurements, laboratory measurements relevant for atmospheric science, and/or theoretical calculations of measurements simulations with detailed error analysis including instrument simulations.*

The development of the CIAO laboratory can be seen as development of measurement instrument and observational techniques. Atmospheric measurements of interest are provided into the paper.

Among journals we selected AMT firstly because the CAIO observatory in its old configuration (only remote sensing part) is described in a previous AMT paper. Secondly, other observatory similar papers appeared on AMT (e.g. doi.org/10.5194/amt-16-6097-2023 and doi:10.5194/amt-8-3481-2015

Comments

page 1, line 26ff: You use the term aerosol here but actually that are the particles or the particulate matter that has these effects. It need to be changed in the whole manuscript.

We will adjust the terminology throughout the manuscript by replacing **aerosol** with **aerosol particle** or simply **particle**, wherever particles are specifically intended. This will ensure greater clarity and precision in conveying the effects attributed to particulate matter.

page 3/4, line 74 and Fig 2: Why to use the radiosonde ground level measurement for RH ant temperature to assess ground level dewpoint temperatures? This can be done with any meteorological equipment with higherdata resolution.

We agree with your point. In the revised version of the paper, Fig. 2 will show the dew point temperature measured by the VAISALA MILOS520 automatic weather station, with daily averages. This weather station was also used to generate the wind rose shown in Fig. 3, ensuring consistency in data sources.

page 4, fig 3: The color scheme in intervals to show also the percentage is a good idea but the color repetition and the polar plot that has its advantage in showing the wind direction makes the reading of low wind speed percentages rather complicated. Btw, the Vaisala MILOS is the data logger of the weather station if I remember well. What anemometer was used?

Thank you for your insightful feedback. We have revised the wind rose to avoid color repetition, ensuring that the plot is now clearer and easier to interpret, especially when reading the percentages for low wind speeds. The updated wind rose diagram is available in the supplemental PDF files under "New Wind Rose Diagram."

Regarding the anemometer installed, it is a mechanical sensor, specifically the Vaisala WA15 model. It consists of the Vaisala WAA151 wind speed sensor and the WAV151 wind direction sensor. These sensors are installed at approximately 10 meters above ground level. We have also included the specifications provided by Vaisala in the supplementary files for reference (wind sensor-wa15).

page 8, section 4: While the inlets are described in high detail, I didn't find the simple parameter of the inlets heights? You tell they are vertical, i.e. rooftop inlets and a height above the roof, but how heigh is that above the ground? Especially as you have a weather station there, mentioned in fig 3.

We will include the height of the inlets from the ground level in the manuscript, not just their height above the roof of the container, to provide a more complete description.

The Vaisala MILOS weather station mentioned in Figure 3, which is used to derive the wind rose diagram and provide an overview of the CIAO observatory, is not located on the in-situ shelter. Instead, it is positioned approximately 50 meters away and at a height of around 10 meters above the ground.

page 9, lines 212ff: You describe here the inlet lines diameters in great detail. In the part where you describe the isokinetic splitter it remains unclear if you have one splitter with several outputs or if you use several splitters one after another? From the description this can be only guessed.

We have two isokinetic splitters with multiple outputs for the various instruments: one is located under the PM10 line, and the other is under the PM2.5 line. This ensures that both aerosol fractions are sampled appropriately for the different instruments. We will clarify this in the manuscript to avoid any ambiguity.

page 9, line 219: Its not clear what you mean with sharp ends here? Their position to be right-angled (90°) in the air stream?

It doesn't need to be at a right angle (90°) in the air stream, but the end should be beveled, not flat. The reason for this is to minimize turbulence and particle loss as the airflow enters the inlet. A beveled edge ensures a smoother transition for the aerosol into the sampling system, which helps maintain the integrity of the sample by reducing potential particle deposition at the edges of the inlet. This also improves the efficiency of the sampling process.

The following revised sentence will be added to the updated version of the paper:

"Moreover, the tube ends in the isokinetic flow splitter must be sharp to minimize turbulence and promote smooth airflow, ensuring uniform sampling. This design helps maintain laminar flow, reduces aerosol losses, and enhances the accuracy and reliability of measurements."

page 9, line 220: The statement of the sampling from the laminar main stream is, at least in engineering, a trivial statement. Do you use an off-the-shelf splitter or was it self made?

The isokinetic flow splitters were custom-built to meet ACTRIS requirements by "4S SOLUZIONI E SVILUPPO PER LA STRUMENTAZIONE SCIENTIFICA," a company specialized in metrology, measurement physics, and the development of scientific instruments, with a particular focus on atmospheric observation tools. In the manuscript, we will include photos to provide a detailed view of the splitters.

page 9/10, table 2: The table layout is a bit awkward which is most probably due to the split over two pages and may be solved by change in the place of the table.

We will adjust the table layout and change its position in the manuscript to ensure that it is not split over two pages, which should resolve the awkward formatting.

page 10/11: Paragraphs on the Nafion dryer system, you discuss the drying capacities, however, did you also determine the losses in the dryer and the whole inlet line in general? E.g. Zoller et al. (2000) report for a rather similar system up to 37% losses on 10nm particles where 20% is lost in the dryer section. Do you have strategies to compensate for inevitable losses in sample lines?

(Zoller, J., Gulden, J., Meyer, J. _et al._ Loss of Nanoparticles in a Particulate Matter Sampling System Applied for Environmental Ultrafine Particle Measurements. _Aerosol Sci Eng_ **4**, 50–63 (2020). https://doi.org/10.1007/s41810-020-00054-6)

In our study, we focused on the drying capacity of the Nafion dryer system but did not specifically measure the particle losses in the dryer or the overall inlet line.

Anyway, the general configuration of the system's inlet line is designed to ensure laminar flow within the sampling line, which is essential for minimizing turbulence and reducing particle loss. Additionally, the use of a conductive tube is kept to a minimum length to further decrease the potential for particle deposition.

Moreover, the system incorporates a Nafion®-based Perma Pure MD-700 air sampling dryer, featuring a large-diameter inlet and a 0.700" diameter flow path. According to test data from TROPOS-WCCAP, this large-diameter Perma Pure MD-700 dryer demonstrates very low particle losses during practical operation, making it an effective solution for maintaining the integrity of particle sampling.

Reference:  https://www.permapure.com/wp-content/uploads/2014/06/MD-700-TROPOS-Presentation-10-2014.pdf

However, it is well recognized that losses, especially for small particles, can be significant in such systems. In the future, we will certainly assess the actual particle loss in the dryer and the entire inlet line. Additionally, we will work to identify the appropriate strategies to compensate for these losses and ensure accurate measurements.

page 16, section 5: Synergistic approach or synergy between measurement systems. The way these are described here is complementary, not synergistic. Two or more measurement systems give details on the same process and each could be used to explain the process. A synergy would create a new aspect that can not be reached by each of the methods alone.

When we talk about synergy, we refer to the ability to reveal new phenomena or insights that emerge only using together the 2 types of observations. Other researchers (e.g. Davulien et al., 2023 doi.org/10.1016/j.scitotenv.2023.167585) intended synergistic approach in a similar way.

Anyhow, based on reviewer's comments we decided to adopt combined instead of synergistic word in the text for what has been done up to now, but leaving the concept of synergistic approach for the further developments that we imagine and plan for the near future, but that are not yet achieved. The current paper is focusing on the implementation of the instruments in view of such synergistic approaches, which will be object of further paper(s).

 Technical remarks

page 13, line 316: TOF-ACSM; I think you already introduced the manufacturer before (page 8) no need for redundant mentioning, maybe check over the manuscipt and as well for other devices.

We will ensure that the manufacturer's name is only mentioned once, as it was already introduced on page 8. We will also check the rest of the manuscript for any other redundant mentions of device manufacturers and streamline the text accordingly.

page 25, line 641: filed = field

It will be done

---

## Author Comment (AC4)

[Figure]

**Figure 3: Wind rose diagram at CIAO in 2004-2017 obtained from continuous measurements of the automatic weather station VAISALA MILOS520 with temporal resolution of 1 minute.**

---

## Author Comment (AC5)

P.O. Box 26, FI-00421 Helsinki, FINLAND
Tel: +358 9 894 91
Fax: +358 9 8949 2485
Email: industrialsales@vaisala.com
www.vaisala.com/WA15

**WA15 Wind Set for High Performance Wind Measurement**

[Figure]

*The WA15 is based on accurate sensors installed on a large crossarm. It is designed for demanding wind measurement applications.*

**Features/Benefits**

- High-performance wind measurement set
- Long and successful track record in meteorological applications
- Accurate wind speed and direction measurement
- Low measurement starting threshold
- Conical anemometer cups provide excellent linearity
- Heated shaft prevents bearings from freezing

With a proven track record of successful installations, the Vaisala Wind Set WA15 has earned its reputation as the industry standard in the wind sensor market.

The WA15 consists of a Vaisala Anemometer WAA151, a Vaisala Wind Vane WAV151, an optional crossarm, a power supply and cabling.

**Anemometer with excellent linearity**

The WAA151 is a fast response, low-threshold anemometer. Three lightweight, conical cups mounted on the cup wheel, provide excellent linearity over the entire operating range, up to 75 m/s.

A wind-rotated chopper disc attached to the shaft of the cup wheel cuts an infrared light beam 14 times per revolution. This generates a pulse output from the phototransistor.

The output pulse rate is directly proportional to wind speed (e.g. 246 Hz = 24.6 m/s). However,  for the highest accuracy, the characteristic transfer function should be used to compensate for starting inertia. (See technical data.)

**Sensitive wind vane**

The WAV151 is a counter-balanced, low-threshold, optoelectronic wind vane. Infrared LEDs and phototransistors are mounted on six orbits on each side of a 6-bit GRAY-coded disc. Turned by the vane, the disc creates changes in the code received by the phototransistors. The output code resolution is ±2.8°.

**Heated bearings withstand cold weather**

Heating elements in the shaft tunnels of both the anemometer and vane keep the bearings above freezing in cold climates.

**Complete package available**

The anemometer and vane are designed to be mounted on Vaisala crossarms.

The WHP151 power supply provides the operating and heating power needed for the WA15. The power supply, as well as the signal and power cables are available as options.

[Figure]

*The WHP151 power supply provides both the operating and heating power needed for the WA15.*

**Technical Data**

**Vaisala Anemometer WAA151**

**Wind speed**

| | |
|---|---|
| Measurement range | 0.4 ... 75 m/s |
| Starting threshold | < 0.5 m/s * |
| Transfer function | U = 0.328 + 0.101 × R |
| (where U = wind speed [m/s], R = o/p pulse rate [Hz]) | |
| Accuracy (within range 0.4 ... 60 m/s) | |
| with characteristic transfer function | ± 0.17 m/s ** |
| with transfer function U = 0.1 × R | ± 0.5 m/s |

**General**

| | |
|---|---|
| Transducer output level | |
| with Iout < +5 mA | high state > $U_{in}$ −1.5 V |
| with Iout > −5 mA | low state < 2.0 V |
| Settling time after power turn-on | < 30 µs |
| Operating power supply | $U_{in}$ = 9.5 ... 15.5 VDC, 20 mA typical |
| Heating power supply | AC or DC 20 V, 500 mA nominal |
| Plug | MIL-C-26482 type |
| Cabling | 6-wire cable through crossarm |
| Recommended connector at cable end | SOURIAU MS3116F10-6P |
| Operating temperature | |
| with shaft heating below +0 °C | −50 ... +55 °C (-58 ... +131 °F) |
| storage temperature | −60 ... +70 °C (-76 ... +158 °F) |
| Material | |
| housing | AlMgSi, grey anodized |
| cups | PA, reinforced with carbon fibre |
| Dimensions | 240 (h) × 90 (Ø) mm |
| Swept radius of cup wheel | 91 mm |
| Weight | 570 g |

**Test compliance**

| | |
|---|---|
| Wind tunnel tests | ASTM standard method D5096-90 |
| ( for starting threshold, distance constant, transfer function) | |
| Exploratory vibration test | MIL-STD-167-1 |
| Humidity test | MIL-STD-810E, Method 507.3 |
| Salt fog test | MIL-STD-810E, Method 509.3 |

Complies with EMC standard EN61326-1:1997 + Am1:1998 + Am2:2001; Generic Environment

* Measured with cup wheel in position least favoured by flow direction. Optimum position gives approx. 0.35 m/s threshold.
** Standard Deviation

**Vaisala Wind Vane WAV151**

**Wind direction**

| | |
|---|---|
| Measurement range at wind speed 0.4 ... 75 m/s | 0 ... 360° |
| Starting threshold | 0.4 m/s |
| Resolution | ±2.8° |
| Damping ratio | 0.19 |
| Overshoot ratio | 0.55 |
| Delay distance | 0.4 m |
| Accuracy | better than ±3° |

**General**

| | |
|---|---|
| Operating power supply | $U_{in}$ = 9.5 ... 15.5 VDC, 20 mA typical |
| Heating power supply | AC or DC 20 V, 500 mA nominal |
| Output code | 6-bit parallel GRAY |
| Output levels | |
| With Iout < +5 mA | high state > $U_{in}$ −1.5 V |
| With Iout > −5 mA | low state < 1.5 V |
| Settling time after power turn-on | < 100 µs |
| Plug | MIL-C-26482 type |
| Cabling | 10-wire cable through crossarm |
| Recommended connector at cable end | SOURIAU MS3116F12-10P |
| Operating temperature | |
| with shaft heating below +0 °C | −50 ... +55 °C (-58 ... +131 °F) |
| storage temperature | −60 ... +70 °C (-76 ... +158 °F) |
| Housing material | AlMgSi, grey anodized |
| Dimensions | 300 (h) × 90 (Ø) mm |
| Swept radius of vane | 172 mm |
| Weight | 660 g |

**Test compliance**

| | |
|---|---|
| Wind tunnel tests | ASTM standard method D 5366-93 |
| ( for starting threshold, distance constant, transfer function) | |
| Exploratory vibration test | MIL-STD-167-1 |
| Humidity test | MIL-STD-810E, Method 507.3 |
| Salt fog test | MIL-STD-810E, Method 509.3 |

Complies with EMC standard EN61326-1:1997 + Am1:1998; Am2:2001; Generic Environment

**Vaisala Wind Set WA15**

**Options and accessories**

| | |
|---|---|
| Crossarm and termination box | WAC151 |
| 16-lead signal cable | ZZ45048 |
| 6-lead power cable | ZZ45049 |
| Crossarm and analog transmitter | WAT12 |
| 6-lead cable for signal and power | ZZ45049 |
| Crossarm and serial RS485 transmitter | WAC155 |
| Serial RS485 transmitter card | WAC155CB |
| Power supply | WHP151 |
| Set of bearings and gasket | 16644WA |
| Cup assembly | 7150WA |

Specifications subject to change without prior notice.
©Vaisala Oyj

CE

---

## Author Response (AR1)

**REFEREE 1**

The above manuscript describes the set-up and characteristics of a combined aerosol in situ and remote sensing measurement side, operated by CNR in ACTRIS. At the beginning, I had my doubts if AMT is the right place for this. But there are, at least for me, two strong arguments, why the manuscript should become an AMT publication. Firstly, for each infrastructure, a reference paper is needed, where, hopefully, all the scientific papers to come can refer to. Secondly, as the authors claim, such a paper can act as "a practical guide for implementation", in particular for researchers in America or Asia, where ACTRIS is probably not so well known. However, to make these two arguments valid, some more detailed information must be provided that the paper can act as reference. And if it should be a practical guide for implementation, for me it is a must to give at least an overview about associated resources (time, man-power, maintenance costs, etc.), both concerning the implementation as well as for the operation later on. This would a with ACTRIS not familiar person allow to do a cost benefit analysis. It would be interesting to go further into that direction and give an estimation on how much of these stations would be needed across Europe or globally to cover the scientific needs. But this is only a nice-to-have remark, no request.

We thank the reviewer for this comment that helped us to improve the paper in order to better underline the importance of the paper for extra European /outside ACTRIS community. Indeed, the ACTRIS aerosol in-situ standards are in some way following the GAW ones, therefore the interest in technical solutions for an ACTRIS compliant in-situ instrumentations stays not only with stations potentially involved in ACTRIS. More relevant for a European perspective, new air quality directive is under definition and this legislation is taking into account standards developed into ACTRIS for example for black carbon (BC) and ultrafine (UF) particles. The approach of such a new EU air quality directive is to have BC and UF measurements in some more advanced stations of the National Air Quality management system. Solutions adopted for collecting such measurements with ACTRIS standard could be of interest for AQMN for guarantying the quality of collected data.

In the revised version of the document, a short sentence has been added about the potential broad interest of our article outside ACTRIS at European and international level:

**Lines 67-74 of manuscript track changes:** A guidance for building-up an ACTRIS aerosol in-situ station, it is potentially of interest also for extra European/outside of ACTRIS community: the ACTRIS in situ standards are in some way following the GAW ones, therefore the interest in technical solutions for an ACTRIS compliant in-situ instrumentations stays not only with the stations potentially involved in ACTRIS. Additionally, new EU air quality directive will include some more advanced stations where black carbon and ultrafine measurements should be collected. Therefore, solutions adopted for collecting such measurements with ACTRIS standard could be of interest for air quality management networks for guarantying the quality of the collected data.

About resources associated to the CIAO aerosol in-situ component, we estimated a total initial investment of about 1 M€, and 2 years were needed for building it up from scratch. For the operation, we estimate about 30k€ as maintenance and 70€ as consumables as annual operational costs. In addition, at least 2 researchers fulltime is needed for running this in-situ instrumentation with the support of a technicians (half time).

Such indications are shortly provided in the revised version of the paper:

**Lines 320-322 of manuscript track changes:** The implementation of a such wide aerosol in situ measurements facility from scratch has required a total initial investment of about 1M€ and about 2 years, and about 100k€, 2 researchers with the support of a technicians are estimated to be needed to operate the laboratory.

How many of such stations are needed to cover the scientific needs: very important question to which is difficult if not impossible to reply to. The geographical density (and the location) of needed number of stations depends on the specific scientific question. Aerosols are highly variable in space and time and therefore measurements collected in different places can strongly differ. For air quality issues, aerosol in-situ measurements should be collected where more inhabitants are for investigating the impact on health, but for decoupling the background levels from extremely local source (e.g. the single car), background stations are needed. If instead the scientific aim is understanding how much the EU policies allow the reduction of PM, few stations maybe just one in remote site could be sufficient.

GAW indeed makes a distinction between: **Local stations** for research and supporting services related to urban environments, and in other locations impacted by nearby emissions (e.g. from biomass burning); **regional stations** for which the station location is chosen such that, for the variables measured, it is regionally representative and is normally free of the influence of significant local pollution sources or at least frequently experiences advection of pollution-free air from specific wind directions; finally **global stations** primarily observe GAW variables under background conditions, i.e. without permanent significant influence from local pollution sources (https://wmoomm-my.sharepoint.com/:w:/g/personal/jbourdeu_wmo_int/EXLkdQpkP-RBi1ooPA3zXpkBHxUFFhX25aJTnj4L4b8k6Q?e=xdra6a). Even if with this classification, the GAW implementation plan is not currently providing a clear directive related to the number of stations and currently GAW accounts for 8 Global stations over Europe, while over Northern America only 2 (https://community.wmo.int/en/activity-areas/gaw/research-infrastructure/gaw-stations/gaw-global-stations) .

In ACTRIS an intermediate solution is adopted, trying to have enough stations in each participating country to cover different regional characteristics. In Italy for example we have aerosol in situ site in Po Valley (north and polluted area), Potenza as background site in the Apennine (south), Lecce Southern Italy a city on the seaside, and Rome as big metropolitan area in the central Italy.

**Specific remarks:**

1. p. 1, l. 13: CNR, IMAA and all the other abbreviations in the manuscript. Please write the full name, when you us the acronym the first time. And as the manuscript has so many acronyms, please add a glossary.

   All abbreviations, including CNR-IMAA, have been spelled out in full the first time they are mentioned in the manuscript. Additionally, we have included a glossary to help clarify all acronyms used in the text (**Appendix A, pages 38-40 of manuscript track changes**).

2. p. 1. l. 18: I´m not so sure if the provided examples are examples for a "synergistic approach", both approaches, remote sensing and in situ, a complementary and hence help each other to get the full picture. My understanding of synergy would be that a combination of these two methods provide a totally new aspect, which cannot be obtained by one method alone. But please convince me that I´m wrong.

   When we talk about synergy, we refer to the ability to reveal new phenomena or insights that emerge only using together the 2 types of observations. Other researchers (e.g. Davulien et al., 2023  doi.org/10.1016/j.scitotenv.2023.167585) intended synergistic approach in a similar way. One example of this type of "synergistic approach" is related to the wintertime aerosol particle conditions. During winter, low clouds and fog often occur at CIAO and therefore lidar

measurements are inhibited or no aerosol optical properties can be retrieved from lidar measurements.

In small number of cases, aerosol particle properties profiles are obtained by lidar measurements in winter.

The climatological profile of aerosol backscatter at 532 nm for winter season 2000-2019 at CIAO (https://doi.org/10.57837/cnr-imaa/ares/actris-earlinet/level3/climatological/2000_2019/pot) shows very clean air respect to other seasons in the whole investigated atmospheric column.

Only the last point close to the surface is slightly higher, but the information content is too low for further investigation. These cases are typically considered as clean day from the aerosol remote sensing perspective.

[Figure]

But it should be considered that the lidar is blind in the lowest portion of the atmosphere and it is expected that due to the low BLH, most pollutant stays within the low BL area.

Aerosol in situ measurements instead do no see above the boundary layer height, but well capture what's happening close the surface. In wintertime the BC is higher at our site probably because of the increase in using heating system (typically fireplaces). Only such in-situ observations allow to understand that winter cases are not to be considered as background conditions below the boundary layer height.

On the other side, remote sensing excels at capturing large-scale, vertical, and temporal variations in aerosol distributions (e.g., the spread of wildfire smoke or desert dust layers), while in-situ instruments are able to offer a detailed chemical composition, size distribution, and ground-level concentrations. This synergy enables scientists to not only track dust but also assess its immediate and long-term impacts on ecosystems and human health, which would be unattainable with either method alone.

The unique value here comes from being able to validate and interpret remote sensing data using in-situ measurements, and vice versa, creating new insights that wouldn't be possible with only one method. For example:

While remote sensing and in-situ approaches are complementary in nature, the true synergy emerges when these methods work together to offer new insights and reduce uncertainties that

cannot be achieved by one alone. This synergy is essential in environmental monitoring, as it helps create more accurate, multi-dimensional views of atmospheric processes, ultimately leading to better scientific understanding and policy decisions. Another aspect of synergy lies in reducing uncertainties in models. Aerosol-climate models, for instance, rely on accurate ground truth data. Remote sensing gives a broader atmospheric overview, but it can suffer from calibration issues or may lack the fine details on chemical composition. In-situ data can calibrate and validate these remote measurements, ensuring that models predict both the spread and chemical impacts of aerosols more accurately. This is a completely new layer of certainty that one method alone cannot provide.

Some revisions have been made on the paper to tackle the reviewer's comment.
**Firstly, based on reviewer's comments we decided to adopt combined instead of synergistic word in the text for what has been done up to now, but leaving the concept of synergistic approach for the further developments that we imagine and plan for the near future, but that are not yet achieved. The current paper is focusing on the implementation of the instruments in view of such synergistic approaches, which will be object of further paper(s).**
**Secondly the cases section will be reviewed (see reply to comment 23).**

3.  p. 1 l. 28: Pöschl, 2005 reference is fine, but two decades old. Maybe add also a newer reference?

The following more recent references have been added to the manuscript to supplement the reference in Pöschl (2005):

**Line 29 of manuscript track changes**: (IPCC, 2021; Ren-Jian et al., 2012)

**Lines 966-968 of manuscript track changes**: IPCC (2021). Aerosols and their impact on climate and human health. In Climate Change 2021: The Physical Science *Basis*. Contribution of Working Group I to the Sixth Assessment Report of the Intergovernmental Panel on Climate Change (IPCC). Cambridge University Press.

**Lines 1101-1102 of manuscript track changes**: Ren-Jian, Z., Kin-Fai, H., & Zhen-Xing, S. (2012). The Role of Aerosol in Climate Change, the Environment, and Human Health. Atmospheric and Oceanic Science Letters, *5*(2), 156–161. https://doi.org/10.1080/16742834.2012.11446983

4.  p. 2 l. 34: I miss the refrence to the ACTRIS BAMS  overvies paper here

The reference to the ACTRIS BAMS overview paper has been added to the manuscript:

**Line 36 of manuscript track changes**: (Laj et al., 2024)

**Lines 971-980 of manuscript track changes**: Laj, P., Myhre, C. L., Riffault, V., Amiridis, V., Fuchs, H., Eleftheriadis, K., Petäjä, T., Salameh, T., Kivekäs, N., Juurola, E$E., Saponaro, G., Philippin, S., Cornacchia, C., Arboledas L. A., Baars, H., Claude, A., De Maziére, M., Dils, B., Dufresne, M., Enamgeliou, N., Favez. O., Fiebig, M., Haeffelin, M., Hermann, H., Höhler, K., Illmann, N., Kreuter, A., Ludewig, E., Marinou, E., Möhler, O., Mona, L., Murberg, L. E., Vicolae, D., Novelli, A., O'Connor, E., Ohneiser, K., Petracca Altieri, R. M. Picquet-Varrault, B., Van Pinxteren, D., Pospichal, B., Putaud, J-P., Reimann, S., Siomos, N., Stachlewska, I., Tillmann, R., Voudori, K. A., Wandinger, U., Wiedensohler, A., Apituley, A., Comerón, A., Gysel-Beer, M., Mihalopoulos N., Nikolova. N., Pietruczuk, A., Sauvage, S., Sciare, J., Skov, H., Svendby, T., Swietlicki, E., Tonev, D., Vaughan, G.,

Zdimal, V., Baltensperger, U., Doussin, J-F., Kulmala, M., Pappalardo, G., Sundet, S. S., Vana, M.: Aerosol, Clouds and Trace Gases Infrastructure (ACTRIS); The European Research Infrastructure Supporting Atmospheric Science, BAMS, E1098-E1136, https://doi.org/10.1175/BAMS-D-23-0064.1, 2024.

5. p. 2 l. 56: the "labelling process" is well known for ACTRIS people. But already European users are unfamiliar with this, not to speak about people from other continents. As the labelling process is an important part of the data quality assurance in ACTRIS, it should be **shortly** described what it is for and how it works, and for the details reference should be given.

I agree with your point, and we have added the following brief description along with a detailed reference:

**Line 58-61 of manuscript track changes**: The site has begun the next phase of the labelling process in 2024, a key element of ACTRIS's data quality assurance system. This process ensures that instruments, data, and methodologies used across ACTRIS observational platforms meet specific quality criteria. The labelling process involves a series of evaluations and certifications to verify compliance with ACTRIS protocols (Deliverable 5.1: ACTRIS NF Labelling Plan).

**Lines 919-921 of manuscript track changes**: Deliverable 5.1: ACTRIS NF Labelling Plan, https://www.actris.eu/sites/default/files/Documents/ACTRIS%20IMP/Deliverables/ACTRIS%20IMP_WP5_D5.1_ACTRIS%20NF%20Labelling%20Plan.pdf

6. p. 3 l. 67: "reference observatory for atmospheric research"? Who claims this? A reference for atmospheric dynamics and ozone hole chemistry? Probably not. I can imagine that it is "a reference station for short-live atmospheric constituents in Italy and the Mediterranean".

You are correct, and I appreciate your insight. The term "reference observatory for atmospheric research" may be too broad and misleading in this context. It would be more accurate to describe it as "a reference station for short-lived atmospheric constituents in Italy and the Mediterranean."

This change will clarify the specific focus of the station and better reflect its role in monitoring atmospheric dynamics and constituents relevant to our studies. Thank you for bringing this to my attention; the modification will help ensure that the description is precise and aligns with the station's actual contributions to atmospheric research.

Done (**Line 84 of manuscript track changes**)

7. p. 3 photo: a lot of infrastructure, all of this is in situ aerosol? Better to zoom in and show the in situ aerosol containers and the inlets.

In the photo, we show the various infrastructures of the CIAO site, which provides a complete view of the facilities. However, we agree that it would be useful to include a zoomed-in photo specifically highlighting the in-situ aerosol containers and entrances. This will provide a clearer understanding of the aerosol sampling setup. We have added the image of the in-situ aerosol shelter to the manuscript (**Line 91 of manuscript track changes**).

8. p. 5 l. 122: you state to list the "research lines" in the following rows here. But "development of", "implementation of", "harmonization of" etc. are no research lines, they are intended steps to allow your research later on. Hence the simplest way to make this consistent would be to replace "research lines" here with a more adequate wording.

In the revised manuscript, the section regarding the "research lines" is no longer present because we have shortened the description of the remote sensing components to highlight the innovative aspect developed at CIAO.

9. p. 6, table 1: the list of instruments is compelling, but, as an in situ person, I would rather like to know which parameters are provided by the remote sensing devices. This would be a good suggestion anyhow, make a table providing all the remote sensing and in situ parameters at the same spot, thus the potential synergy gets more visible.

In the revised manuscript, we have updated the table to include both the remote sensing and in situ instruments, along with the respective parameters they measure in the **Section 5, Table 2, pages 25-26-27 of manuscript track changes**) . This will help to better highlight the potential synergy between the different measur-ement -techniques.

10. P-. 6 l. 14-4: "Observational Platform" and "Exploratory Platform" are known to some of us, but not to the waste majority of the readers. Please explain shortly (in parentheses) what these terms stand for.

In the revised manuscript, the terms "Observational Platform" and "Exploratory Platform" are no longer present because we have shortened the description of the remote sensing components to highlight the innovative aspect developed at CIAO.

11. p. 6 l. 152: The description of the Central Facility part of CIAO is surely correct, but not needed for the purpose of the paper and rather confusing for the reader. Please omit this here, it is better described elsewhere.

We have condensed the description of the Central Facility part of CIAO in this section, as it is not essential for the purpose of the paper and may cause confusion (**Section 3**).

12. p. 9. L 211.: which heads do the PMx instruments have? Please give this information.

This information is already provided in section 4.4, which details the instrument configuration. However, to clarify it here as well:

**Line 246-249 of manuscript track changes**: Additionally, two PMx samplers (SWAM 5a-Dual Channel Monitors, FAI Instruments) are installed with respective inlets: one equipped with two PM2.5 inlets, and the other with one PM10 and one PM1 inlet. Furthermore, a PMx monitor (EDM 180, Grimm) is placed as a standalone instrument with individual PM10 inlet line.

13. p. 9 l. 218: the isokinetic flow splitter: which one did you use? Or can you provide as drawing of it?

The isokinetic flow splitters were custom-built to meet ACTRIS requirements by "4S SOLUZIONI E SVILUPPO PER LA STRUMENTAZIONE SCIENTIFICA," a company specialized in metrology, measurement physics, and the development of scientific instruments, with a particular focus on atmospheric observation tools. In the manuscript, we have included a photo to provide a detailed view of the splitters (**Line 266, Figure 5 of manuscript track changes**).

14. p. 9 l. 219: I have some experience with sampling lines, but I only can guess the argument of the sharp tube ends, please be more specific.

The ends of the tube in the isokinetic flow splitter must be sharp to ensure a homogeneous distribution of the air sample because sharp edges help minimize flow disturbances and

turbulence at the entrance of the sampling tubes. A sharp edge promotes a smooth transition of the air flow, reducing the risk of vortices and irregular flow patterns that can lead to uneven sampling.

When the air enters the tubes with a well-defined, sharp edge, it helps maintain the laminar flow conditions that are crucial for accurately capturing the aerosol particles. This ensures that the sampled air maintains a consistent velocity and composition, allowing for representative measurements and reducing the potential for aerosol losses due to inertial effects or diffusion. In essence, a sharp tube end facilitates better mixing and uniformity in the airflow, which is vital for the accuracy and reliability of aerosol measurements.

The following revised sentence has been added to the revised version of the paper:

**Line 259-261 of manuscript track changes**: Moreover, the tube ends in the isokinetic flow splitter must be sharp to minimize turbulence and promote smooth airflow, ensuring uniform sampling. This design helps maintain laminar flow, reduces aerosol losses, and enhances the accuracy and reliability of measurements.

15. p. 9 table 2: first of all, the two first Reynolds numbers are equal, even if the flow rate is different. The upper one is wrong in my opinion. Same for the speed there.

There was indeed a typo error in the table regarding the flow rate of the aethalometer, **which is not 3 l/min but 5 l/min**. As a result, both the aethalometer and the nephelometer, having the same internal diameter of the isokinetic splitter and flow rate, also have the same Reynolds number and flow speed. This has been corrected in the manuscript.

16. p. 10 l. 228: for me one of the most critical points. Knowing from the literature and also from own experience, conductive "plastic" tubes can be critical, both concerning particle losses as well as chemical composition. The chosen tube MIGHT be OK, but please either provide a reference for that or provide own measurement data e. g. on the size-resolved particle transmission. Otherwise all your data are always "conditionally" correct only.

We sincerely thank the reviewer for highlighting this critical point. In the manuscript, there was an incorrect definition regarding the material of the tubes used. The tubes employed are actually TSI sampling black tubes.

The revised manuscript contains the following new description of the tubes used:

**Line 275-280 of manuscript track changes**: In addition, in accordance with ACTRIS recommendations, the tubes used are black sampling tubes supplied by TSI company (https://tsi.com/home/). These TSI sampling tubes are made of conductive silicone, infused with carbon black to improve conductivity. This design is essential to minimize electrostatic losses, which can occur in non-conductive tubes, such as those made of standard silicone or Teflon, where particles can adhere to the tube walls due to static charges. The conductive nature of TSI tubes prevents the buildup of electrostatic fields, thus improving particle penetration and reducing sampling distortions caused by particle loss.

17. p. 10 l. 232: the Nafion dryer, which one? Please provide reference or explain, what this is

A Nafion dryer is a specialized device used in aerosol sampling to remove water vapor from a gas stream while preserving the integrity of the sample's chemical composition. Nafion membrane is a sulfonated tetrafluoroethylene-based polymer that selectively transfers water vapor from a gas sample to a surrounding purge gas, while retaining the sample's other gases and particles. This property makes it ideal for removing moisture from aerosol-laden air without affecting the aerosol particles themselves.

We have included the following explanation and reference in the manuscript regarding the Nafion dryers we use:

**Line 284-287 of manuscript track changes:** In compliance with the ACTRIS indications, all the instruments in the laboratory are equipped with a Nafion dryer tube, a specialized device made from a sulfonated tetrafluoroethylene-based polymer. This device is used in aerosol sampling to remove water vapor from the gas stream while preserving the chemical integrity of aerosol particles (Monotube Dryer 700 (MD-700) - Perma Pure).

18. p. 10 l. 240: why is the 2:1 flow ratio desired, please explain

The 2:1 purge-to-sample flow ratio in a Nafion dryer operate in reflux mode is crucial for achieving efficient aerosol particulate sampling. This ratio ensures there is enough dry purge gas to continuously absorb moisture from the sample, which prevents the purge from becoming saturated. By maintaining this higher purge flow, the system can keep moisture levels low, ensuring that the sample's integrity is preserved. This is particularly important for aerosol particulate sampling, where even small amounts of moisture can alter the particle characteristics and lead to inaccurate measurements. The 2:1 ratio helps maintain consistent drying efficiency over time, which is essential for reliable aerosol analysis.

Here is the revised version that we included in the manuscript with the shortened explanation:

**Line 295-299 of manuscript track changes**: The vacuum on the purge air should be at least 15 inches Hg, with a higher vacuum preferable. This vacuum level is required to provide the desired 2:1 purge-to-sample flow ratio based on the actual volumetric flow. The 2:1 ratio ensures enough dry purge gas to continuously absorb moisture, preventing saturation and preserving sample integrity. This is crucial in aerosol particulate sampling, where even small amounts of moisture can affect particle characteristics and compromise measurement accuracy.

19. p. 13 l. 317: "unattended measurements ... on the timescale of years" is an overstatement, you have to check the instruments regularly, even if they might be OK for one or the other year (which I personally doubt). Please soften this statement.

We agree that the statement could be softened. While the robustness of the instrument was the point being emphasized, it is indeed essential that the instruments are regularly checked and calibrated to ensure the accuracy and reliability of long-term measurements. We have modified the wording to reflect this and avoid the implication that the instruments can operate indefinitely without maintenance:

**Line 379-382 of manuscript track changes:** For what concerns the aerosol particles mass spectrometry techniques, the ToF-ACSM (Aerodyne Research) has been shown to be perfectly suited for the ACTRIS observatory platforms. It is specifically designed to provide continuous aerosol particle monitoring over long time periods, spanning years, with the requirement of

regular checks and calibrations to maintain the accuracy and reliability of its long-term measurements.

20. p. 13 l. 338: the statement that Potenza is a rural site is a trivial statement, please remove or phrase differently, what you want to highlight

We have removed the statement referring to Potenza as a rural site, as it might be too simplistic.

**Line 405-407 of manuscript track changes**: As a general comment, we could say that the Potenza site exhibits low PM concentrations and a very high contribution of the organic substances, as observed in rural areas.

p. 15 l. 366: in the manuscript, I miss some more evaluation data, checking the consistence of the measurements e.g. here, how good the mass measurements of the different instruments agree with the size distribution derived mass etc.

The reviewer is perfectly right. Several tests can be made for cross checking the instruments (between the aerosol in-situ instruments as well as versus the aerosol remote sensing ones). Anyhow the aim of the current paper is the description of the different steps and solution adopted for implementing such a large aerosol in-situ laboratory coupled to an existing remote sensing observatory. Part of the checks suggested by the reviewer are part of the quality assurance procedures working in ACTRIS and we are proceeding with those. Additional cross checks will be done taking the most from the plethora of CIAO instruments and expertise. However long record of data would be desirable for an assessment of the consistency and accuracy of the measurements. The results will be included and fully explained in subsequent manuscripts.
A short sentence about this will be added in Final section of the revised paper.

**Line 61-64 of manuscript track changes:** During the labelling process, the National Facilities are annually invited by CAIS - ECAC (Center for Aerosol In-Situ – European - Center for Aerosol Calibration and Characterization) to calibration workshops, where instruments are calibrated, and the quality of the data is thoroughly verified.

21. p. 15 l. 370ff: The elemental analysis, what is given in ACTRIS there or is this just an add-on to the in situ aerosol particle properties?

The elemental analysis of aerosol particles is a recommended variable by ACTRIS, though it is not mandatory for a site to become an ACTRIS National Facility (NF) observatory. Despite not being compulsory, elemental analysis of in situ aerosol particles plays a crucial role in understanding the composition and sources of atmospheric aerosols, as well as their impact on air quality, health, and climate.

The elemental analysis provides important data on the concentration of potentially toxic elements and trace metals in particulate matter, helping to distinguish between natural and anthropogenic sources. This type of analysis is essential for more detailed studies on the environmental and health impacts of aerosols.

For this reason, we have equipped our site with ICP-OES (Inductively Coupled Plasma Optical Emission Spectroscopy) and an OC/EC analyzer (Organic Carbon/Elemental Carbon) to perform comprehensive elemental and carbon content analysis, which significantly enhances our capability to assess aerosol properties in detail.

In the revied paper we add a short sentence at beginning of section 4.1 about mandatory and recommended instruments for ACTRIS aerosol in situ facility.

**Line 228-233 of manuscript track changes:** This facility enables the measurement of all obligatory ACTRIS aerosol in-situ variables: particle number concentration > 10 nm; particle number size distribution – mobility diameter 10 to 800 nm; particle light scattering & backscattering coefficient and particle light absorption coefficient and equivalent black carbon concentration. Additionally, it allows the measurement of other four recommended variables: particle number size distribution - aerodynamic diameter 0.8 to 10 μm; mass concentration of particulate organic and elemental carbon; mass concentration of non-refractory particulate organics and inorganics and mass concentration of particulate elements.

22. p. 17 l. 433: the three cases: I believe I understood what the authors wanted to show here, but I have the feeling that most of the statements can be already given with only one of the two methods, in situ and remote. I do not see the real synergy. This would be the case, at least for me, if you would use in situ and remote sensing data to generate another data product. Please elaborate a little bit more on that section, otherwise you weaken your own argument that collocated measurements are valuable. At the same time please shorten the section, there is too much "text book" knowledge in.

This comment is linked to the point 2 above, where we discussed the general framework. Here the comment is more specific to the three show cases. As reported in the paper, "we present three emblematic cases recurring at CIAO where the combined deployment of the in- situ and remote sensing observations is expected to be of added value". Here we describe the 3 topics on which we expect the most from the, combined first and synergistic then, use of remote sensing and in-situ data. The description of these topics allows also to report first 10 months record of black carbon measurements collected at CIAO and a show case of desert dust arrival over CIAO captured by the lidar and photometer measurements and the identification of its intrusion down to the ground by aerosol in-situ measurements.

To address the reviewer comment, for the three cases we shorten the section by reducing the textbook-like knowledge and focus more on the added value of combining these measurement techniques. In particular: 1) local wild fire subsection has been mentioned as potential case of investigation in the text (**Section 5.1**) ; 2) winter pollution: paucity of aerosol remote sensing, low BLH and blind lidar region for this period and needs of independent and in-situ measurements for such period relevant for air quality (health) issue will be underlined (**Section 5.2**); 3) desert dust: it will be better underlined how such case demonstrates firstly the agreement between the 2 observations in capturing a desert dust event, underlining how each one of the technique overcomes the limit of the other one and finally we will describe potential future synergistic products (**Section 5.3**).

Moreover, I believe modelling could strongly benefit from collocated in situ and remote sensing measurements, but this is not addressed in the manuscript.

This suggestion has been addressed in the revised version of the manuscript by highlighting the added value of the availability of in situ collocated measurements and remote sensing of aerosols in increasing the accuracy of model predictions, allowing the reduction of uncertainty of aerosol measurements in the atmosphere (e.g., Vratolis et al., 2020), as well as in the evaluation of aerosol models. Furthermore, in recent years, the use of collocated aerosol measurements has found application in training machine learning-based models

In the revised version, short sentences on this aspect and on the relevance of in situ aerosol measurements to qualify the findings of microphysical properties based on remote sensing measurements have been explicitly reported:

**Line 504-508 of manuscript track changes:** Furthermore, the availability of collocated in situ and remote sensing measurements of aerosols also represents an added value for modelling. Indeed, it can contribute to the increase in the accuracy of model predictions, allowing the reduction of the uncertainty of aerosol measurements in the atmosphere (e.g., Vratolis et al., 2020), as well as to a better evaluation of aerosol models. In recent years, collocated datasets have also been increasingly utilized for training machine learning-based models, as demonstrated by Redemann and Gao (2024).

References:

**Line 1099-1100 of manuscript track changes:**: Redemann, J. and Gao, L.: A machine learning paradigm for observations needed to reduce uncertainties in aerosol climate forcing, Nature Communications, 15, 8343, 2024.

**Line 1145-1148 of manuscript track changes:**: Vratolis, S., Fetfatzis, P., Argyrouli, A., Soupiona, O., Mylonaki, M., Maroufidis, J., Kalogridis, A.-C., Manousakas, M., Bezantakos, S., Binietoglou, I., Labzovskii, L.D., Solomos, S., Papayannis, A., Močnik, G., O'Connor, E., Müller, D., Tzanis, C.G., Eleftheriadis, K.: Comparison and complementary use of in situ and remote sensing aerosol measurements in the Athens metropolitan area, Atmos. Environment, 228, 117439, 2020.

23. p. 24. l. 620: the conclusion is not a conclusion, it is mainly written as outlook. Outlook is fine in at the end of the conclusion, but should not cover the major part of the text. Hence please rephrase the conclusions. Therefore imagine, e.g. the three most important statements the reader should have learned reading your paper.

The conclusions in the revised manuscript have been modified as follows, highlighting the most important statements the reader should have learned from reading the paper (**Section 6**):

"The recent upgrade of aerosol in-situ laboratory to the well-established remote sensing activities at the CIAO observatory significantly enhances its observational capacity. The integration of in-situ and remote sensing measurements offers a more complete understanding of aerosol behaviour, enabling detailed studies from ground level up to the stratosphere. This combination adds value by providing both vertical profiles by remote sensing measurements and precise ground-level chemical and physical properties through in-situ measurements, which is crucial for improving climate models and understanding aerosol impacts on human health.

Establishing the aerosol in-situ facility has been a complex and labour-intensive endeavour. The process, which began in 2018, required careful planning, technical expertise, and collaboration with field specialists. The setup involved designing and implementing ACTRIS-compliant inlets, sampling lines, and advanced instruments to ensure accurate and reliable measurements. This development highlights the significant effort required to meet international standards and provide high-quality data for the scientific community.

Given CIAO's strategic location in the Mediterranean, the case studies planned for future research are especially relevant. The site is frequently affected by Saharan dust intrusions, which impact air quality and ecosystems, and the observatory is strategically positioned to study these phenomena. Moreover, the Mediterranean is also prone to wildfires, which are projected to increase in intensity and frequency due to climate change. The CIAO observatory can monitor both the short-range transport of smoke from local fires and long-range plumes from major events, providing insights into their effects on air quality and human health. Lastly, local winter pollution, which results from residential heating, can also be analysed in detail, particularly during temperature inversions that trap pollutants near the ground. The combination of in-situ and remote sensing measurements will help investigate these key environmental issues.

Furthermore, the next-to-come ICOS Atmospheric Class 1 site at CIAO (first step of labelled process already passed) will offer other possibilities of synergistic studies and integration among Ris in the environmental filed. In this direction, CIAO is deeply involved in the developments of ITINERIS (Italian Integrated Environmental Research Infrastructures System), an overarching National project for enhancing the interlinkages of all the Italian Ris in the environmental domain. The multi-platform and multi-disciplinary approach of the observatory coupled with the open data and open access philosophy is key for better addressing complex atmospheric and environmental questions posed by climate change and anthropization processes."

**Technical corrections:**

1.  p. 1, title: isn´t it "building-up" with a hyphen?

    Done

2.  p. 1 l. 21: not sure if "container organization" does fit in here, "container layout" sounds better. Also "optimization" should be deleted in l. 22 in order to make the list more homogeneous.

    Done

3.  p. 1 l. 26: might sound nitpicking but I believe it is important to state "aerosol particle" or "particle" everywhere, where the particles are meant, and not "aerosol", which are the particles and the surrounding gas. Please check this in the whole manuscript.

    The term aerosol has been changed to "aerosol particle" or simply "particle" throughout the manuscript where specific reference is made to particles, ensuring clarity and precision in the terminology used.

4.  p. 2 l. 33: Please add "The" before "Aerosol Clouds and Tr...."

    Done

5.  p. 2 l. 45: here CNR-IMAA has a hyphen, on page one not. Please be consistent.

    Done

6.  p. 4 wind rose: you might have used all the wind speed classes shown in the legend, but in practical, only winds up to 10 m/ show up. Hence make the higher wind speed classes just one additional " and larger" bin. This also prevents that the same color shows up more than once in the legend.

Thank you for your valuable suggestion. We fully agree with your observation, and the wind rose has been adjusted accordingly. The higher wind speed classes have been consolidated into one additional "and larger" bin to reflect the practical wind speeds observed, preventing the repetition of colours in the legend (**Line 112 of manuscript track changes**).

7. p. 5 l. 95: either "Small and Medium-sized Enterprises" or "small and medium-sized enterprises" but not a mixture of both small and capital letters.

   Done

8. p. 5 l. 98: delete the "and", because there still follows the "or" in the list

   Done

9. p. 5 l. 98: please make it either "to contribute instruments," or "to contribute to the instrumentation,"

   Done

10. p. 5 l. 100: please exchange "revolve around" with "evolved within"

    Done

11. p. 5 l. 104: please exchange "measurements" with "data", because you provide the quality-assured data for the satellite validation.

    Done

12. p. 5 l. 114: please exchange "smokes" with "smoke plumes"

    Done

13. p. 6 l. 128: please add an "and " before " i)"

    Done

14. p. 6, table 1.: "lidar and optical laboratories" are infrastructure, no "instrument" as stated in the table caption.

15. p. 6 l. 149: I learned that there should always be a space between the number and the unit, i. e. should be "20 km" here. (Only exception "10°C"). Please check the whole manuscript.

    The entire manuscript was reviewed and we ensured that there was a space between the number and the unit (e.g. "20 km"), except in specific cases such as "10°C". All cases were corrected accordingly.

16. p. 7 l. 152: should be "Central Facilities" starting with capital letters. Please check the whole manuscript.

    All instances have been corrected accordingly.

17. p. 8 l. 199: please make it either "particulate matter collected on filters." or "aerosol particles collected on filters."

    Done

18. p. 8 Fig. 4 caption: please add "aerosol" before "facility"

    Done

19. p. 9 l. 206/209: please exchange "under" with "downstream", because that is meant

Done

20. p. 9 l. 208: please move the comma after "3938" and remove the hyphen to be consistent

Done

21. p. 9 Fig. 4: please add the information of the inlet heads also to the PMx instruments

Done

22. p. 10 l. 244: please exchange "Instead" with "In contrast"

Done

23. p. 10 l. 244: please move the "since …" half sentence to the end of the sentence, the subject and the verb should not be separated.

Done

24. p. 11 l. 250: please exchange "input" with "inflow"

Done

25. p. 11 l. 259: please add "matter" after "particulate".

Done

26. p. 12 l. 281: "human range of visibility" sounds strange for me, maybe "the visible part of the electromagnetic spectrum" is better

Done

27. p. 13 l. 322: what is " < ng m$^{-3}$", please complete this equation or write it in words

"< ng m$^{-3}$" was written in words (e.g., "less than nanograms per cubic meter"). This change was made in the manuscript.

28. p. 14 l. 340: please exchange "increasing" with "peaks"

Done

29. p. 14 Fig. 7: please add "a)" and "b)" to the two rows of plots and change the figure caption accordingly

Done

30. p. 16 l. 415: "aerosol load" is very unspecific, please use a more appropriated term for what is meant here

the term "aerosol load" has been replaced with the more specific term "aerosol concentration"

31. p. 17 l. 446: please exchange "in short" with "over short"

Done

32. p. 20 l. 530/532: please remove the "%" after "BB", same in fig. 8 b

we do not remove "%" as it refers to the percentage of BC created by biomass burning, determined by the Sandradewi model and hence to BB%

33. p. 21 l. 553: should be "coarse" not "carse"

Done

34. p. 25 l. 649: why some words in the "authors contribution" section are written in capital letters and others not, is not clear to me.

The words in the "Authors' Contributions" section were capitalized to highlight the different contributions of the authors, in line with the guidelines provided by the AMT journal. However, we have changed this setting to all lowercase letters.

35. p. 25 l. 662, acknowledgement: Most of the guidelines for setting-up an in situ aerosol side in ACTRIS are given by the in situ aerosol Topical Centre. Hence, here and also in the text this TC should be acknowledged, as many years of hard work are the basis for this.

Done

**REFEREE 2**

The paper describes the CIAO laboratory that have been recently added an aerosol in-situ measurement component that is aimed to add to the ACTRIS RI in future.

Overall, I am a bit worried if the paper is publishable in it's current shape. On the one hand there is the title stating the in-situ upgrade. That is described but ther eis as well a large portion of general description of the station that have in part already described by some papers of some of the authors.

The processes that are linked to the compliance with ACTRIS and the procedures to achive that, like "labelling" are a very specific in frame of European research infrastructures and play a minor role outside these communities. Therefore, such terms are kind of a jargon and needed some explanation, at least.

We thank the reviewer for his comment that pushes us to improve our paper. We have shortened the description of the remote sensing components to highlight the innovative aspect developed at CIAO (**Section 3**).

We have also given an explanation about the ACTRIS jargon in particular for the term "labelling" to make it easier for non-ACTRIS readers to read it (**Line 58-64**).

The paper tries to fulfil AMT's requirements but that leads to a situation where the scientific strenght remains low and the bare description of a measurement station or system alone without innovation may not qualify for the journal. However, I can imagine that AMT may set rules to make descriptions of such large scale efforts and stations or systems and by that reduce the antagonistic problem these type of descriptions are causing.

The aim of our paper is **to benefit the aerosol community providing a comprehensive and detailed description of technical solutions for the implementation of such component** (aerosol in situ). The development of such laboratory for aerosol in situ measurements required designing implementing and optimizing technical solutions. This paper is the way in which we want to share our experience allowing others to do not repeat all the process but adopting and in case of needs tailoring our solutions to their needs. Examples of measurements and their combined use with aerosol remote sensing observations are reported as show cases of potentialities of CIAO extended observatory.

Such a topic perfectly fits in our opinion with AMT scope: *The main subject areas comprise the development, intercomparison, and validation of measurement instruments and techniques of data processing and information retrieval for gases, aerosols, and clouds. Papers submitted to AMT must contain atmospheric measurements, laboratory measurements relevant for atmospheric science, and/or theoretical calculations of measurements simulations with detailed error analysis including instrument simulations.*

The development of the CIAO laboratory can be seen as development of measurement instrument and observational techniques. Atmospheric measurements of interest are provided into the paper.

Among journals we selected AMT firstly because the CAIO observatory in its old configuration (only remote sensing part) is described in a previous AMT paper. Secondly, other observatory similar papers appeared on AMT (e.g. doi.org/10.5194/amt-16-6097-2023 and doi:10.5194/amt-8-3481-2015

Comments

page 1, line 26ff: You use the term aerosol here but actually that are the particles or the particulate matter that has these effects. It need to be changed in the whole manuscript.

The term aerosol has been changed to "aerosol particle" or simply "particle" throughout the manuscript where specific reference is made to particles, ensuring clarity and precision in the terminology used.

page 3/4, line 74 and Fig 2: Why to use the radiosonde ground level measurement for RH ant temperature to assess ground level dewpoint temperatures? This can be done with any meteorological equipment with higherdata resolution.

We agree with your point. In the revised version of the paper, Fig. 2 shows the dew point temperature measured by the VAISALA MILOS520 automatic weather station, with daily averages. This weather station was also used to generate the wind rose shown in Fig. 3, ensuring consistency in data sources (**Line 103, Figure 2 of manuscript track changes**).

page 4, fig 3: The color scheme in intervals to show also the percentage is a good idea but the color repetition and the polar plot that has its advantage in showing the wind direction makes the reading of low wind speed percentages rather complicated. Btw, the Vaisala MILOS is the data logger of the weather station if I remember well. What anemometer was used?

Thank you for your insightful feedback. We have revised the wind rose to avoid colour repetition, ensuring that the plot is now clearer and easier to interpret, especially when reading the percentages for low wind speeds (**Line 112, Figure 3 of manuscript track changes**).

Regarding the anemometer installed, it is a mechanical sensor, specifically the Vaisala WA15 model. It consists of the Vaisala WAA151 wind speed sensor and the WAV151 wind direction sensor. These sensors are installed at approximately 10 meters above ground level.

page 8, section 4: While the inlets are described in high detail, I didn't find the simple parameter of the inlets heights? You tell they are vertical, i.e. rooftop inlets and a height above the roof, but how heigh is that above the ground? Especially as you have a weather station there, mentioned in fig 3.

We have included the height of the inlets from the ground level in the manuscript, not just their height above the roof of the container, to provide a more complete description.

**Line 280-283 of manuscript track changes:** The inlets on the rooftop of the field laboratory are placed one metre apart and positioned at a height of 1.5–2.0 m above the roof, corresponding to approximately 4.5–5 m above ground level, with the aim of minimising local influences and potential interferences in the sampling process.

The Vaisala MILOS weather station mentioned in Figure 3, which is used to derive the wind rose diagram and provide an overview of the CIAO observatory, is not located on the in-situ shelter. Instead, it is positioned approximately 50 meters away and at a height of around 10 meters above the ground.

page 9, lines 212ff: You describe here the inlet lines diameters in great detail. In the part where you describe the isokinetic splitter it remains unclear if you have one splitter with several outputs or if you use several splitters one after another? From the description this can be only guessed.

We have two isokinetic splitters with multiple outputs for the various instruments: one is located under the $PM_{10}$ line, and the other is under the $PM_{2.5}$ line. This ensures that both aerosol fractions are sampled appropriately for the different instruments. We have clarified this in the manuscript to avoid any ambiguity:

**Line 255-257 of manuscript track changes:** The instrument sublines (characterised by smaller inside diameters) are connected to the two common $PM_{10}$ and $PM_{2.5}$ inlets through their respective isokinetic flow splitters (Figure 5), where the sample flow velocity closely matches the velocity of the main flow.

page 9, line 219: Its not clear what you mean with sharp ends here? Their position to be right-angled (90°) in the air stream?

It doesn't need to be at a right angle (90°) in the air stream, but the end should be blunt, not flat. The reason for this is to minimize turbulence and particle loss as the airflow enters the inlet. A blunt edge ensures a

smoother transition for the aerosol into the sampling system, which helps maintain the integrity of the sample by reducing potential particle deposition at the edges of the inlet. This also improves the efficiency of the sampling process.

The following revised sentence has been added to the revised version of the paper:

**Line 259-261 of manuscript track changes**: Moreover, the tube ends in the isokinetic flow splitter must be sharp to minimize turbulence and promote smooth airflow, ensuring uniform sampling. This design helps maintain laminar flow, reduces aerosol losses, and enhances the accuracy and reliability of measurements.

page 9, line 220: The statement of the sampling from the laminar main stream is, at least in engineering, a trivial statement. Do you use an off-the-shelf splitter or was it self made?

The isokinetic flow splitters were custom-built to meet ACTRIS requirements by "4S SOLUZIONI E SVILUPPO PER LA STRUMENTAZIONE SCIENTIFICA," a company specialized in metrology, measurement physics, and the development of scientific instruments, with a particular focus on atmospheric observation tools. In the manuscript, we have included a photo to provide a detailed view of the splitters (**Line 265, Figure 5 of manuscript track changes**).

page 9/10, table 2: The table layout is a bit awkward which is most probably due to the split over two pages and may be solved by change in the place of the table.

the table will be formatted in the final version of the paper

page 10/11: Paragraphs on the Nafion dryer system, you discuss the drying capacities, however, did you also determine the losses in the dryer and the whole inlet line in general? E.g. Zoller et al. (2000) report for a rather similar system up to 37% losses on 10nm particles where 20% is lost in the dryer section. Do you have strategies to compensate for inevitable losses in sample lines?

(Zoller, J., Gulden, J., Meyer, J. _et al._ Loss of Nanoparticles in a Particulate Matter Sampling System Applied for Environmental Ultrafine Particle Measurements. _Aerosol Sci Eng_ **4**, 50–63 (2020). https://doi.org/10.1007/s41810-020-00054-6)

In our study, we focused on the drying capacity of the Nafion dryer system but did not specifically measure the particle losses in the dryer or the overall inlet line.

Anyway, the general configuration of the system's inlet line is designed to ensure laminar flow within the sampling line, which is essential for minimizing turbulence and reducing particle loss. Additionally, the use of a conductive tube is kept to a minimum length to further decrease the potential for particle deposition.

Moreover, the system incorporates a Nafion®-based Perma Pure MD-700 air sampling dryer, featuring a large-diameter inlet and a 0.700" diameter flow path. According to test data from TROPOS-WCCAP, this large-diameter Perma Pure MD-700 dryer demonstrates very low particle losses during practical operation, making it an effective solution for maintaining the integrity of particle sampling.

Reference: https://www.permapure.com/wp-content/uploads/2014/06/MD-700-TROPOS-Presentation-10-2014.pdf

However, it is well recognized that losses, especially for small particles, can be significant in such systems. In the future, we will certainly assess the actual particle loss in the dryer and the entire inlet line. Additionally, we will work to identify the appropriate strategies to compensate for these losses and ensure accurate measurements.

page 16, section 5: Synergistic approach or synergy between measurement systems. The way these are described here is complementary, not synergistic. Two or more measurement systems give details on the

same process and each could be used to explain the process. A synergy would create a new aspect that can not be reached by each of the methods alone.

When we talk about synergy, we refer to the ability to reveal new phenomena or insights that emerge only using together the 2 types of observations. Other researchers (e.g. Davulien et al., 2023 doi.org/10.1016/j.scitotenv.2023.167585) intended synergistic approach in a similar way.

Anyhow, based on reviewer's comments we decided to adopt combined instead of synergistic word in the text for what has been done up to now, but leaving the concept of synergistic approach for the further developments that we imagine and plan for the near future, but that are not yet achieved. The current paper is focusing on the implementation of the instruments in view of such synergistic approaches, which will be object of further paper(s).

 Technical remarks

page 13, line 316: TOF-ACSM; I think you already introduced the manufacturer before (page 8) no need for redundant mentioning, maybe check over the manuscipt and as well for other devices.

We ensured that the manufacturer's name is mentioned only once, as it was already introduced on page 8. Additionally, we reviewed the rest of the manuscript to eliminate any redundant references to device manufacturers.

page 25, line 641: filed = field

Done